# Allosteric activation of vinculin by talin

Florian Franz[1,2,9], Rafael Tapia-Rojo[3,4,9] ✉, Sabina Winograd-Katz [5], Rajaa Boujemaa-Paterski [6], Wenhong Li[5], Tamar Unger[7], Shira Albeck[7], Camilo Aponte-Santamaria [1,2], Sergi Garcia-Manyes [3,4], Ohad Medalia [6] ✉, Benjamin Geiger [5] ✉ & Frauke Gräter [1,2,8] ✉

The talin-vinculin axis is a key mechanosensing component of cellular focal adhesions. How talin and vinculin respond to forces and regulate one another remains unclear. By combining single-molecule magnetic tweezers experiments, Molecular Dynamics simulations, actin-bundling assays, and adhesion assembly experiments in live cells, we here describe a two-ways allosteric network within vinculin as a regulator of the talin-vinculin interaction. We directly observe a maturation process of vinculin upon talin binding, which reinforces the binding to talin at a rate of 0.03 s⁻¹. This allosteric transition can compete with force-induced dissociation of vinculin from talin only at forces up to 10 pN. Mimicking the allosteric activation by mutation yields a vinculin molecule that bundles actin and localizes to focal adhesions in a force-independent manner. Hence, the allosteric switch confines talin-vinculin interactions and focal adhesion build-up to intermediate force levels. The 'allosteric vinculin mutant' is a valuable molecular tool to further dissect the mechanical and biochemical signalling circuits at focal adhesions and elsewhere.

Tissue cells can sense, decode and respond to mechanical cues that depend on the physical and mechanical properties of their immediate surroundings, such as density, topography, and stiffness[1,2]. Focal adhesions play a major role in the mechano-sensing capabilities of these cells as they develop tiny, dot-shaped, and short-lived nascent adhesions inside the lamellipodia[3]. Myosin-II-driven contractility grows these nascent adhesions into mature focal adhesions (FAs) and triggers hierarchical recruitment of FA proteins which form large and complex cytoskeletal structures[4,5]. The core regulatory action of these force-sensitive signaling hubs depends on an intricate interplay between talin, vinculin, and F-actin[3].

Mechanical force plays a pivotal role in all stages of focal adhesions development and turnover. The forces within the cell – generated to a large extent by the actomyosin machinery – can fluctuate strongly in location and time. Interestingly, the locations that exhibit the highest cellular forces have been identified to often coincide with the assembly sites of FAs[6,7]. Tension sensors indicated that talin and vinculin experience forces as high as 10 pN when engaged in FAs[8–10]. Within FAs, vinculin acts as a mechanical reinforcement in the link between the actin cytoskeleton and talin, which in turn is directly bound to the integrin receptors connecting to the cell´s exterior[4,10,11].

Vinculin, a 5-domain protein composed of 1066 residues, is present in all force-bearing cellular junctions such as FAs, adherens junctions, and immunological synapses[12–14]. As an interaction hub and versatile signaling molecule, vinculin harbors binding sites for a variety of adhesome proteins present in these distinct junctional systems[15]. In

[1]Heidelberg Institute for Theoretical Studies (HITS), Schloß-Wolfsbrunnenweg 35, 69118 Heidelberg, Germany. [2]Interdisciplinary Center for Scientific Computing (IWR), Heidelberg University, Mathematikon, INF 205, 69120 Heidelberg, Germany. [3]Department of Physics, Randall Centre for Cell and Molecular Biophysics, Centre for the Physical Science of Life and London Centre for Nanotechnology, King's College London, Strand WC2R 2LS London, UK. [4]Single Molecule Mechanobiology Laboratory, The Francis Crick Institute, 1 Midland Road, London, NW1 1AT London, UK. [5]Department of Immunology and Regenerative Biology, Weizmann Institute of Science, Rehovot, Israel. [6]Department of Biochemistry, University of Zurich, 8057 Zurich, Switzerland. [7]The Dana and Yossie Hollander Center for Structural Proteomics, Weizmann Institute of Science, Rehovot, Israel. [8]IMSEAM, Heidelberg University, INF 225, 69120 Heidelberg, Germany. [9]These authors contributed equally: Florian Franz, Rafael Tapia-Rojo. ✉e-mail: rafael.rojo@kcl.ac.uk; omedalia@bioc.uzh.ch; benny.geiger@weizmann.ac.il; frauke.graeter@h-its.org

its auto-inhibited conformation, vinculin's tail domain (Vt) packs onto its head domain (Vh), and most of its binding sites are cryptic. For vinculin to unfold its full signaling potential, activation is required[16]. However, the exact process by which vinculin is activated is not yet fully understood.

The head domain Vh comprises four-helix bundles, D1-D4, with the D1 domain harboring the interaction site for talin as well as for α-actinin, and α-catenin. Talin's so-called vinculin binding sites (VBS) are single helical domains, which are packed up in helix bundles and masked if mechanical force is absent[17,18]. Talin contains eleven VBS; α-actinin and α-catenin only contain one VBS[19–21]. The structural mechanism by which VBS binds to vinculin is conserved across all vinculin binding partners[20,22,23]. X-ray crystallography structure determination of D1-Vt with and without a bound VBS revealed a mechanism in which VBS binding reorganizes the helix bundle of D1 such that vinculin is partially activated[23].

The three known actin-binding sites on the vinculin tail are either partially or fully occluded in the auto-inhibited conformation, and, hence, actin is not able to bind to inactive vinculin[24–26]. The binding of actin to full-length vinculin requires an artificially weakened head–tail interface[27] or VBS binding[28]. Bois et al.[20]. suggested that the α-actinin VBS alone can cause vinculin activation, while other studies propose a collaborative effort of a VBS and actin[29,30]. Further players in the multimodal activation scenario of vinculin include mechanical force[31] and PIP2[32,33].

While mechanical force across the talin backbone has been unequivocally shown in single-molecule studies to be decisive for VBS exposure and vinculin binding[34–36], a direct role of force for vinculin activation is less apparent. A collision-induced semi-open vinculin state with an increased cross-section area was recently captured using ion-mobility mass spectrometry[37], possibly mimicking a force-induced activated state of vinculin. Interestingly, decreased cellular traction forces increase the unbinding of zyxin from FAs but not of vinculin[38]. Recent findings by Boujemaa-Paterski et al.[28]. revealed an enhanced actin-bundling by vinculin in a force-independent manner, that is, which takes place when full-length vinculin is exposed to a talin VBS1 domain. These findings support the hypothesis that VBS-binding activates vinculin—being it fully or partially—and establish an order of events in which VBS binding proceeds actin binding and thus force transmission onto vinculin.

To test this hypothesis, we used a multidisciplinary integrative approach comprising Molecular Dynamics (MD) simulations, single-molecule force spectroscopy experiments, functional in vitro assays, and live cell imaging approaches. With these tools, we revealed a two-way allosteric mechanism in full atomistic detail of how vinculin regulates talin binding and vice versa. On this basis, we rationally designed vinculin mutants that aim to mimic the underlying destablization in the head–tail auto-inhibitory interface by VBS. For our designed vinculin mutants that simulate the VBS effect on vinculin, we tested actin bundling in vitro and focal adhesion formation in live cultured cells, both in a largely force-independent fashion. Our study disentangles the key steps of vinculin activation and identifies an extensive rewiring of a salt-bridge network as the key mechanism in weakening vinculin's auto-inhibitory head–tail interface. It also puts forward our vinculin mutants as close mimics of vinculins bound to stretched talin and α-actinin, and proposes them as useful molecular tools to study and interfere with cell-matrix and cell–cell adhesion.

## Results

### Vinculin's head-to-tail interaction weakens vinculin-talin binding

Vinculin can bind talin both in its active and autoinhibited state. But whether vinculin in its active open state binds to talin any different than its inactive closed state is unknown. Thus, we investigated the impact of vinculin's head-to-tail interactions on the binding of

stretched talin by force-probe MD simulations. We subjected VBS1 to a set of different constant forces in the range of 200-260 pN (Fig. 1A–C) while being bound to either full-length vinculin (FLV) or only to vinculin's D1 domain (Vd1). We started from a VBS1-vinculin complex (see Methods) and monitored the progression of VBS helix force-induced unraveling by the increase of end-to-end distance (Fig. 1B, Fig. S1). We then deduced Mean First Passage Times (MFPT) for the transition to the first partially unfolded intermediate of VBS1 (Fig. 1C). Across the force regime covered in the simulations, VBS1 shows, on average, shorter MFPTs (Fig. S1D, E) by approximately one order of magnitude when bound to FLV compared to Vd1, while the force-dependence of these times is comparable (Bell-Evans fits[39,40] in Fig. 1C). This suggests that the presence of the head-to-tail interaction destabilizes the complex, leading to a faster loss of helicity and vinculin-talin interactions under force. The atomistic detail of the simulations allowed the identification of a range of residues within the vinculin D1 domain that VBS1 and the tail compete for when bound to vinculin (Fig. S2), explaining this allosteric mechanism.

We next tested this prediction experimentally. To measure the interaction between FLV and talin under force, we purified the proteins and employed single-molecule magnetic tweezers to directly detect individual vinculin-binding events in single talin molecules subjected to forces in the pN range. Previously, we had used magnetic tweezers to characterize the force-dependent binding of Vd1 to the talin R3 domain[35]. These experiments showed that force regulates binding of Vd1 in a biphasic way: force first favors binding by unfolding talin and exposing the cryptic VBSs, but, above 20 pN, this interaction is hampered as the VBSs recoiling transition triggered upon binding (which also provides a single-molecule fingerprint for detecting individual vinculin binding events) becomes energetically unfavorable. Based on this assay, we here investigated binding of FLV to single talin R3 domains subjected to pN-level forces. In our experiments, the talin R3[IVVI] domain—the R3 domain harboring the IVVI mutation, which increases its mechanical stability without impacting vinculin binding except for increasing the force threshold for binding from ~5 to ~8 pN[35,41,42]—is flanked by two mechanically stiff Ig32 domains, serving as molecular handles (Fig. 1D). At the N-terminus, a HaloTag allows specific and covalent anchoring to a glass cover slide, while a biotinylated C-terminus AviTag binds to streptavidin-coated M270 superparamagnetic beads, establishing a highly specific and stable single-molecule tether. By controlling a magnetic field using a magnetic tape head[43,44], we can subject single R3 domains to low mechanical forces (0-40 pN) and monitor their conformational dynamics over physiologically relevant time scales. (Fig. 1D).

In the presence of 20 nM FLV, our experiments show that FLV binds talin R3 similarly as Vd1, contracting the unfolded talin polypeptide by ~3 nm and locking talin in the unfolded state (Fig. 1E, red arrow), which suggests that both FLV and Vd1 bind talin through an analogous mechanism. However, while at low forces—e.g. 8.5 pN, where talin folds and unfolds in equilibrium—no apparent differences between FLV or Vd1 binding are noticed (Fig. 1E, upper panel), when we explored higher forces such as 12.5 pN or 15 pN (Fig. 1E, middle and lower panel), we observed that FLV unbinds after just a few seconds, showing an upwards ~3 nm step that accounts for the uncoiling of the VBSs and the recovery of talin's unfolded extension (Fig. 1E, blue arrows). This is different from the case of Vd1, which forms a highly stable complex, requiring forces in excess of 40 pN to trigger vinculin unbinding over seconds-long timescales[35].

To characterize the nanomechanics of the R3-FLV complex and compare them to those previously determined for Vd1, we measured the force-dependent binding (Fig. 1F) and unbinding (Fig. 1G) kinetics of FLV to talin R3[IVVI]. Interestingly, the binding rates are equivalent to those previously measured for Vd1, which suggests that the binding reaction (on-rate) is dominated by the vinculin D1 domain alone, with no or little influence of the rest of the vinculin head domains (D2–D4),

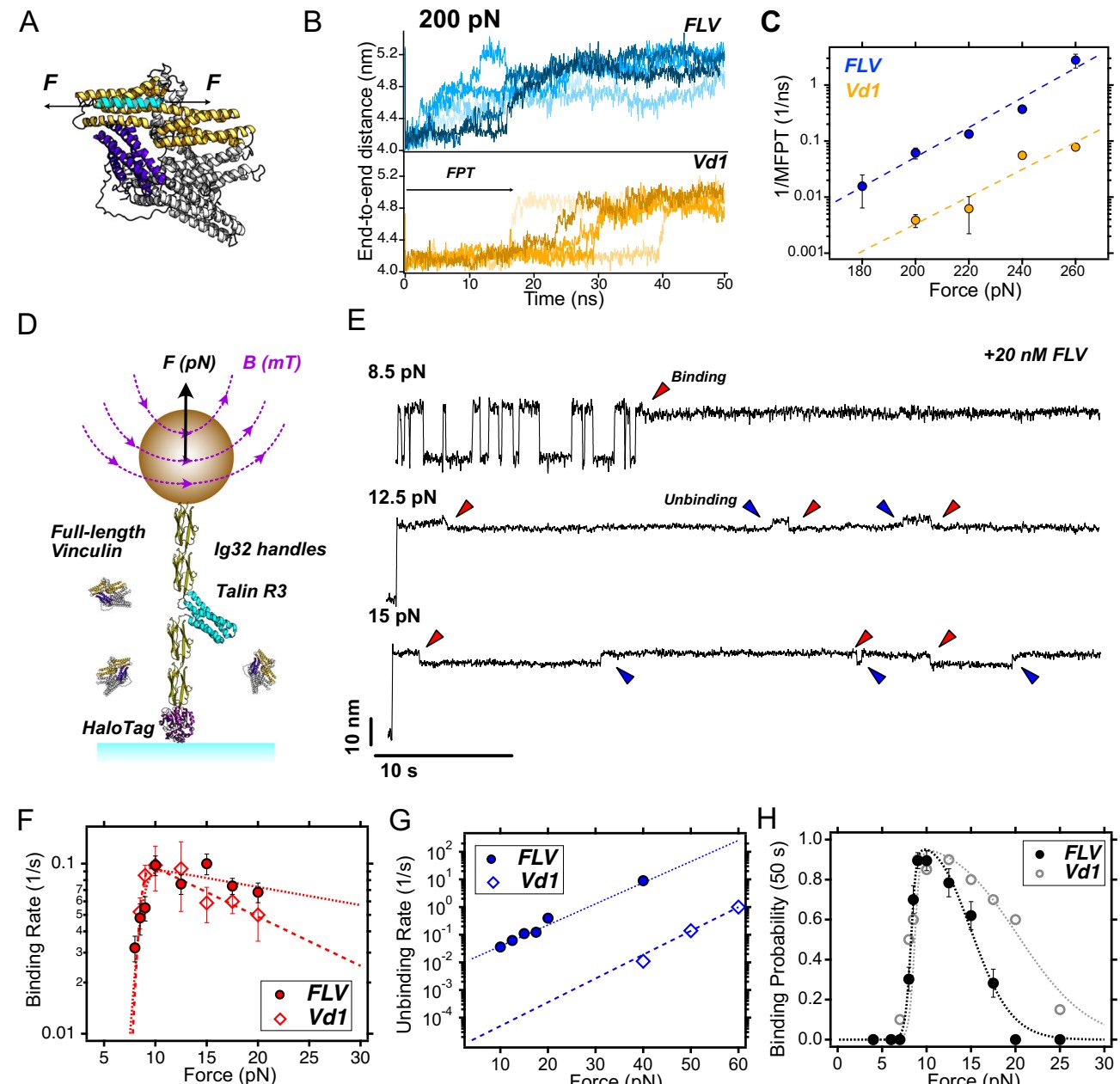

**Fig. 1 | Vinculin's head−tail interface regulates vinculin-VBS binding.**
**A** Structural scheme of full-length vinculin (FLV) binding to a VBS under force (cyan helix). D1 domain (Vd1) is shown in orange, and the tail domain in purple. **B** End-to-end length of VBS1 at 200 pN bound to full-length vinculin (upper panel) and vinculin D1 domain (lower panel) as observed in MD simulations. The increase in extension from 4.0 nm to 5.2 nm corresponds to the first uncoiling transition of VBS1, used as a proxy for vinculin unbinding kinetics (first passage time, FPT). Data for other forces are shown in Fig. S1. **C** Expected rates (inverse MFPT, computed as shown in Figs S1D, E) to first transition as a function of force for FLV (blue) and Vd1 domain (orange). Data fitted to a Bell-Evans's model (FLV: $k_0 = 2.6 \times 10^{-7}$ s$^{-1}$, $x^{\dagger} = 0.25$ nm; Vd1: $k_0 = 4.5 \times 10^{-8}$ s$^{-1}$, $x^{\dagger} = 0.23$ nm). Error bars are SEM. Data from $N = 75$ independent simulations. **D** Schematics of our single-molecule magnetic tweezers assay to measure FLV binding to talin under force. **E** Magnetic tweezers trajectories of R3$^{IVVI}$ in the presence of 20 nM FLV. At 8.5 pN (upper panel), talin folds and

unfolds in equilibrium, and individual FLV binding events can be detected by a ~3 nm contraction of the talin polypeptide (red arrow) and the arrest in talin folding dynamics. At higher forces (12.5 pN, middle panel, and 15 pN, lower panel), the complex is less stable, and reversible binding (red) and unbinding (blue) events are observed as downward and upward ~3 nm steps, respectively. **F** Binding rates of FLV and Vd1 to R3$^{IVVI}$. Data fitted to the Bell-Evans model assuming a positive force dependence for exposing the VBS and a negative force dependence for binding (see Methods). **G** Unbinding rates of FLV and Vd1. Data fitted to the Bell-Evans model (FLV: $k_0 = 6.6 \times 10^{-3}$ s$^{-1}$, $x^{\dagger} = 0.72$ nm; Vd1: $k_0 = 6.8 \times 10^{-6}$ s$^{-1}$, $x^{\dagger} = 0.81$ nm). **H** Binding probability of FLV and Vd1 measured over a 50 s time window. Data for FLV (**F**−**H**) from $N = 725$ FLV binding and unbinding events on 15 talin molecules. Error bars are SEM in all cases. Data for Vd1 adapted from ref. 35. Source data are provided as a Source Data file.

or the tail domain. The biphasic force dependence of the FLV binding rates can be modeled assuming that the binding rates to the exposed VBSs decrease with force following the Bell-Evans model while their exposure follows R3's unfolding kinetics (see Methods and SI for details on the model and fitting parameters). By contrast, the

unbinding rates of FLV are very different from those measured for Vd1. Similar to the behavior observed in the MD simulations (Fig. 1C), the slope of the force dependence is very similar in both cases (FLV: $x^{\dagger} = 0.72$ nm vs. Vd1: $x^{\dagger} = 0.81$ nm), which indicates a comparable distance to the transition state, likely dominated by the uncoiling of the VBSs.

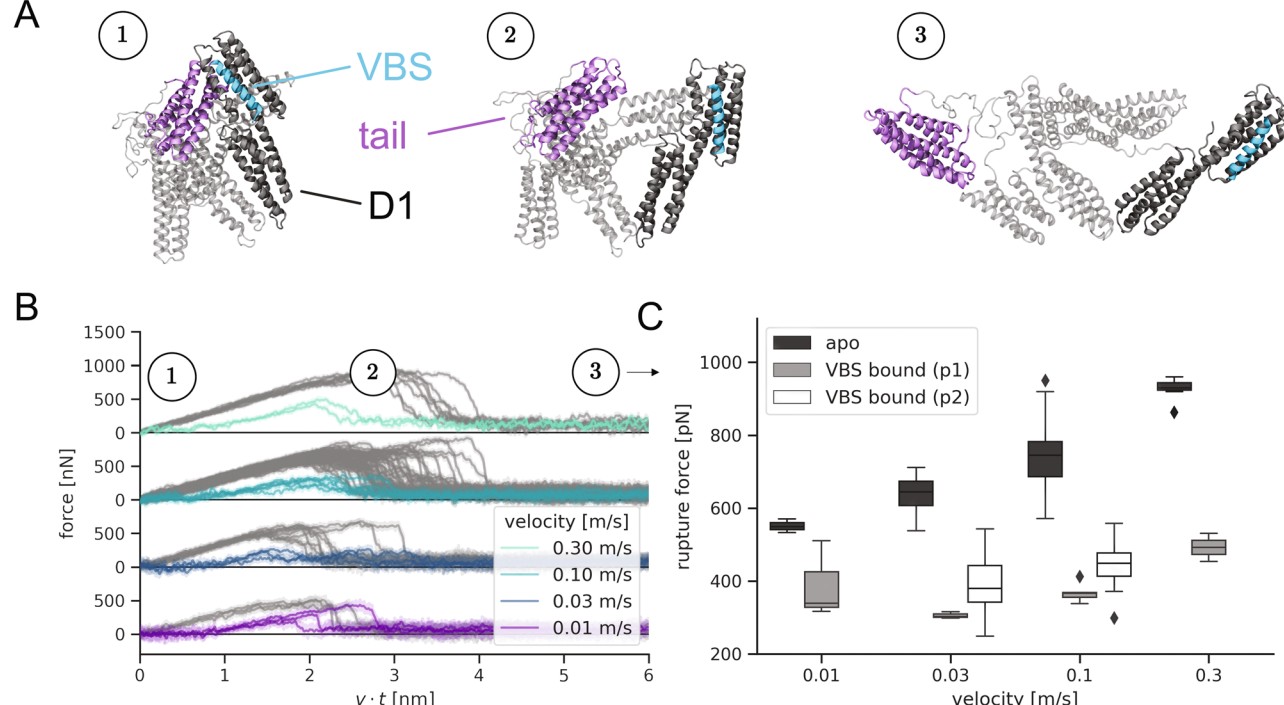

**Fig. 2 | Talin binding facilitates vinculin head–tail opening. A** Snapshots extracted from an exemplary trajectory to represent the opening transition during the force-probe MD simulations (see Fig. 1B and Methods). Force was applied directly to the VBS peptide (cyan) or to the talin binding site located on the vinculin D1 domain (dark gray). All simulations included full-length vinculin (Vh in gray, Vt in purple). **B** Force-extension traces for force application to VBS1 and pulling speeds between 0.01 and 0.3 m/s. As a reference, the force-extension curves for opening of the apo-state protein are shown in gray. **C** Quantification of the recorded rupture forces for the apo-state. For the VBS-complex rupture forces decreased drastically and two points of force application were investigated: (p1) on vinculin at the location of the talin binding site (light gray), which ensued slightly higher rupture forces, and (p2) directly on the VBS (black), which lead to the smallest observed rupture forces. Boxes show the quartiles of the data set, with whiskers extending to account for the rest of the distribution except for outliers. Source data are provided as a Source Data file.

However, the intrinsic off-rates differ by three orders of magnitude (FLV: $k_0 = 6.6 \times 10^{-3}\,\mathrm{s}^{-1}$, and Vd1: $k_0 = 6.8 \times 10^{-6}\,\mathrm{s}^{-1}$), suggesting that FLV forms a much less stable complex with talin R3 than the vinculin D1 domain. When characterizing the equilibrium binding properties as the binding probability measured over a 50 s time window (Fig. 1H), FLV shows a narrower force range for binding.

To demonstrate that this behavior is not specific to R3[IVVI] but intrinsic to FLV, we measured binding of FLV to the R3[WT] domain. Compared to the R3[IVVI] mutant, R3[WT] has lower mechanical stability (-5 pN) and shows rapid folding dynamics (-0.1 s)[41], which makes it harder to characterize from a nanomechanical perspective. Still, we were able to resolve individual vinculin binding events to R3[WT] and observed the same behavior as for FLV binding to R3[IVVI]: FLV unbinds shortly after binding at forces >12.5 pN, suggesting that it forms a much weaker complex than Vd1 (Fig. S4). The only difference between FLV binding to R3[WT] or R3[IVVI] is the lower threshold force for binding (5 pN for R3[WT] versus 8 pN for R3[IVVI]) directly related to its lower unfolding force (Fig. S4), similar to what was previously reported for Vd1[35].

Overall, our MD simulations and single-molecule experiments together indicate that FLV forms a less stable complex with talin than the isolated D1 domain, suggesting that, while auto-inhibited vinculin is still capable of binding talin under force, the head–tail vinculin interaction renders a weaker complex that unbinds after only a few seconds. A stable complex to be formed likely requires vinculin activation by the release of the head–tail interaction, which we investigated next.

## VBS binding facilitates vinculin activation in MD simulations

As suggested by crystallographic data, VBS binding causes major structural changes in the vinculin D1 domain[23]. To reveal the implications of these changes on the strength of the D1-tail interface in full-length vinculin and to rationally design mutants to interfere with these changes, we subjected the equilibrated VBS1-vinculin complex and the apo-state protein to force-probe MD simulations. We now employed an external force between the head and tail domain to enhance the sampling and provoke a dissociation along the most likely pathway of vinculin activation on the time scale of the MD simulations. Figure 2 depicts the simulation system of the VBS1-vinculin complex, with snapshots extracted from the trajectories (snapshots for the apo-state of vinculin are shown in Fig. S3 for comparison).

As expected, the tail domain (purple in Fig. 2A) detaches from the head domain (dark gray). All helical bundles remain intact, and the VBS1 peptide (cyan) stays stably bound to D1. The simulation protocol thus successfully samples the transition towards an activated vinculin conformation. The force-extension curves (Fig. 2B) reveal a pronounced drop in the force required for detaching the tail from the remainder of the protein bound to VBS1. In the apo-state of vinculin, where VBS1 is absent, force-extension curves reach pronouncedly higher rupture forces for opening the head–tail interface. Figure 2C directly compares the rupture forces as a measure for the strength of the auto-inhibitory head–tail interface across four different pulling velocities. VBS1 binding consistently lowers, that is, roughly halves, the rupture force for vinculin activation, e.g. from -550pN to -350pN at the lowest pulling velocity of 0.01 m/s. Additional simulations excluded that the observed drop in rupture forces is caused by the slight difference in pulling direction (pulling from VBS helix versus talin binding site, for details, see Fig. 2C and Methods). To ensure comparability, we limited subsequent analysis and simulations to the case in which force is consistently applied to the talin binding site.

To further validate that the pronounced drop in resistance against activation of vinculin is solely due to the presence and entailed conformational change of VBS and does not arise from other differences in the crystallographic starting structures, we initiated an additional set of simulations from an apo state obtained by deletion of VBS1 from the complex. Remarkably, we observed both the recovery of the D1 conformation close to the experimentally determined apo state as well as of the high activation forces characteristic for this state (Fig. S5).

## Talin binding induces a partial dissociation of vinculin's head–tail interaction

The computational evidence suggests that talin binding facilitates vinculin activation by allosterically weakening the head–tail interaction. Therefore, we wondered whether such conformational transition in vinculin upon binding to talin under force could be captured with our single-molecule approach. While in our magnetic tweezers experiments, we monitor the conformational dynamics of talin and not of vinculin, our experiments strongly suggest that the head–tail interaction weakens the interaction of vinculin with the VBSs under force. Thus, it is enticing to hypothesize that if vinculin activation occurs upon binding to talin as an opening (or at least a partial opening) of the head–tail interface, such conformational change could be indirectly detected at the single-molecule level by an increase in the stability of the talin-vinculin complex under force.

When measuring R3 dynamics at low forces (~8.5 pN) in the presence of FLV over several hours, we observed that vinculin unbinds over a timescale of an ~hour (Fig. 3A). These data indicate that, under physiological forces, FLV forms a stable complex with R3, compatible with the lifetime of focal adhesions[45,46]. However, these slow unbinding kinetics are not consistent with those shown in Fig. 1G, which, extrapolated to forces around 8.5 pN, imply much faster unbinding rates (~0.01 s$^{-1}$). Thus, this seemingly contradictory evidence suggests the existence of two distinct bound states with different stability. To test this hypothesis, we designed a force protocol to examine the stability of the R3-FLV complex (Fig. 3B). First, we allowed FLV bind talin at 8.5 pN and kept this low force during a variable time, to then apply a high-force 40 pN pulse to trigger the mechanical dissociation of vinculin and interrupt the complex. By applying the high-force pulse at different times after the binding event is observed, we characterized the stability of the complex as a function of its lifetime. When we applied the 40 pN pulse soon after vinculin binds, we observed a very rapid dissociation in the sub-second timescale (Fig. 3B, left panel, inset); however, if the high-force pulse is applied several seconds after the R3-FLV complex is formed, the vinculin unbinding kinetics are much slower, taking a few seconds to dissociate (Fig. 3B, right panel). To corroborate the existence of these two different bound modes, we calculated the distribution of unbinding times $p(t_{Ub})$ at 40 pN using a logarithmic binning (square-root histogram, see Methods), which identifies each involved timescale as a peak in the distribution. This method has been successfully implemented to analyze ion-channel recordings with multiple kinetic processes or to detect ephemeral molten globule states in protein folding[44,47]. The distribution shows two peaks, representing a first weak binding mode that quickly dissociates over ~0.4 s, and a stronger one that unbinds over ~7 s (Fig. 3C). To determine whether the existence of these two bound modes was indeed related to the lifetime of the complex—perhaps involving a maturation process to render a tighter interaction—we plotted the unbinding time as a function of the complex lifetime (Fig. 3D). These data show a sigmoidal dependence, indicating that over very short timescales the interaction is characterized by a low stability (compatible with the unbinding rates measured in Fig. 1G), while after a few seconds, the complex evolves towards a more stable interaction, compatible with the slow unbinding rates measured at 8.5 pN (Fig. 3A). From this sigmoidal dependence, we can estimate the timescale for this maturation transition to be ~37 s. This maturation from an initial

weak binding mode to a stronger complex was also observed in FLV binding to R3$^{WT}$, suggesting that this behavior is related to the dynamics of the bound FLV, and not to the nanomechanics of the stretched talin domain (Fig. S6).

Given that Vd1 forms a much more stable interaction with R3, also involving just a single bound state, we hypothesized that the observed maturation involves some conformational transition in FLV, perhaps related to vinculin activation. To this aim, we measured the binding of full-length vinculin head (D1-D4 domains) to R3, which strikingly showed the same binding and unbinding properties as Vd1, also involving a single bound mode (Fig. 3E and Fig. S7). This suggests that, from the four domains that compose the vinculin head, only the D1 domain plays a significant role in interacting with talin under force, also indicating that the maturation process observed for FLV is related to the dynamics of its head–tail interaction and, therefore to vinculin activation. However, the dissociation kinetics of the mature state of FLV are still faster than those of vinculin head or Vd1, implying that the influence of the head–tail vinculin interaction still persists in this mature state. If we would assume that active vinculin would bind talin as strongly as its head alone (no influence of the tail), this would mean that VBS binding is not sufficient to fully activate vinculin, perhaps requiring the participation of other partners such as actin.

Thus, based on our single-molecule data, we can propose a kinetic model to describe the interplay between mechanical forces, vinculin binding, and activation (Fig. 3F). At low forces (up to ~9-10 pN), FLV binds the stretched VBSs with very high affinity ($K_d$ ~ 0.2 nM). Upon binding, vinculin undergoes a conformational transition (likely a partial opening of the head–tail interface) over a timescale of ~37 s, which renders a much more stable complex ($K_d$ ~ 0.004 nM). However, if talin is under higher forces above ~10 pN (still within the physiological range), although vinculin still binds quickly and with high affinity ($K_d$ ~ 5 nM), the unbinding rates are too fast to allow vinculin maturation, rendering a weak complex that dissociates shortly after it forms. The weakened Vd-Vh interface upon VBS binding and its stabilization when removing VBS observed in MD simulations (Fig. 1C and Fig. S5) support this model.

## Rewiring of salt bridge network causes vinculin activation upon talin binding

Talin binding to vinculin strongly undermines vinculin's resistance against the opening of the auto-inhibitory head–tail interface, as shown by our MD simulations and magnetic tweezers experiments. While the comparison of the apo and VBS1-bound states already highlighted the role of electrostatic interactions[23], the MD simulations now provide the full dynamic picture in atomistic detail and allow us to pinpoint the key residue interactions allosterically affected by VBS binding. We calculated the major correlated motions present at the interface of Vh (residues 1–200) and Vt (880–1066) in the tensed state (Fig. 4A and Methods). The apo state shows several clusters of strongly correlated interface residues with significantly diminished correlation upon VBS1 binding (Fig. 4B). The residue pairs with the highest loss of correlation within each cluster (Table S1) are visualized as spheres in Fig. 4C which also displays the networks of increased and diminished interactions as identified by force distribution analysis (see methods section). The overlap in results of the two analysis methods (one based on non-equilibrium simulations, the other on equilibrium simulations) allowed us to locate three areas of strong coupling: residues K944 and R945, as well as residues D1013 and E1015 give away most of their coupling to the regions between the tail residues 19-33, and 88-116. In addition, tail residue E986 loses interactions with a region around residue 185. On this basis, we designed two novel mutants, K944A-R945A-D1013A-E1015A (4M), and an additional mutation E986A (5M), in which the charged residues in the head domain are mutated to alanines in order to abolish the stabilizing salt bridge network, thereby mimicking the VBS-bound matured state. We tested if the allosteric

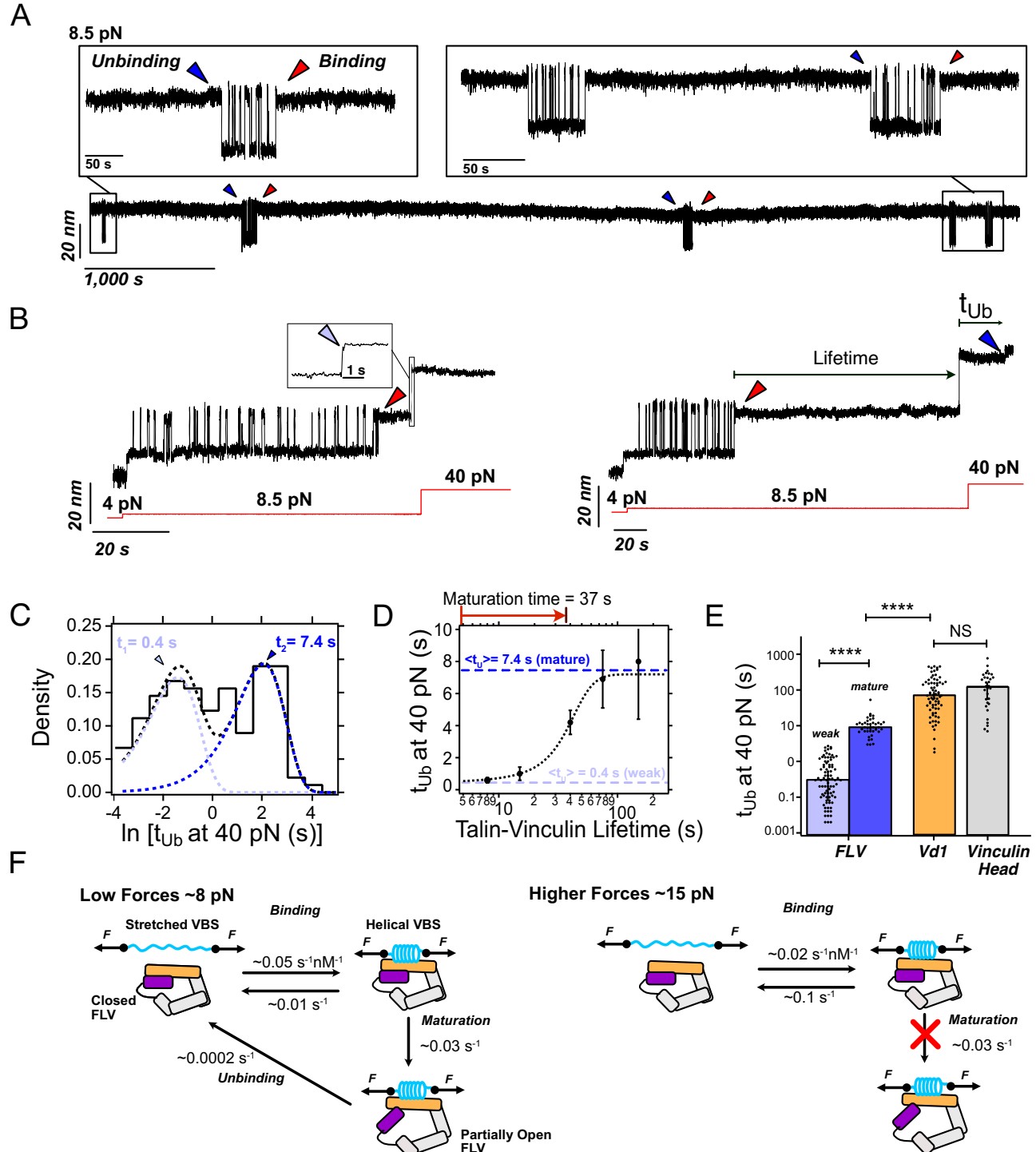

**Fig. 3 | Talin binding weakens the vinculin head-to-tail interaction. A** Magnetic tweezers recording showing the binding dynamics of full-length vinculin to talin R3[IVVI] at low forces over hours-long timescales. At 8.5 pN, FLV can remain bound to talin over ~1 h, indicating much slower unbinding kinetics than those corresponding to higher forces. **B** Magnetic tweezers recordings where the talin-vinculin complex is interrupted by a 40 pN force pulse after being bound for just a few seconds (left) or several seconds (right). Over short lifetimes, FLV unbinds on a sub-second timescale, while if the complex is left to mature at low forces, the interaction is reinforced, and it unbinds at 40 pN after a few seconds, indicating a much more stable complex. **C** Square-root histogram of unbinding times ($t_{Ub}$) at 40 pN. The distribution of unbinding times calculated with logarithmic binning shows two peaks, indicating two unbinding timescales, a fast one of ~0.4 s ($t_1$) and a slower one of ~7.4 s ($t_2$), suggesting two different binding modes. Data from $N = 165$ vinculin unbinding events measured on 12 talin molecules. **D** Unbinding time of FLV at 40

pN plotted as a function of the complex lifetime. The talin-vinculin interaction matures towards a more stable complex over a timescale of ~37 s. Error bars are SEM. **E** Unbinding kinetics of FLV (weak state, light blue; mature state, dark blue), the vinculin D1 domain (orange), and full-length vinculin head (gray). Data from $N = 85$ (weak), $N = 42$ (strong); $N = 74$ (D1), and $N = 32$ (FLVH) vinculin unbinding events. Bars indicate the average unbinding time, and error bars are SD. Significance levels from non-parametric Mann–Whitney test, $NS\ P > 0.05$; ****$P < 10^{-4}$. (**F**) Kinetic diagram suggesting the binding mechanism of FLV to a VBS low force (left) and higher forces (right). At forces in the ~8 pN range, the interaction is reinforced over ~37 s, likely due to a partial opening of vinculin that stabilizes the talin-vinculin complex. At higher forces around ~15 pN, the maturation step is kinetically unfavorable due to the faster competing unbinding kinetics, and FLV only binds VBS in its weak mode. Source data are provided as a Source Data file.

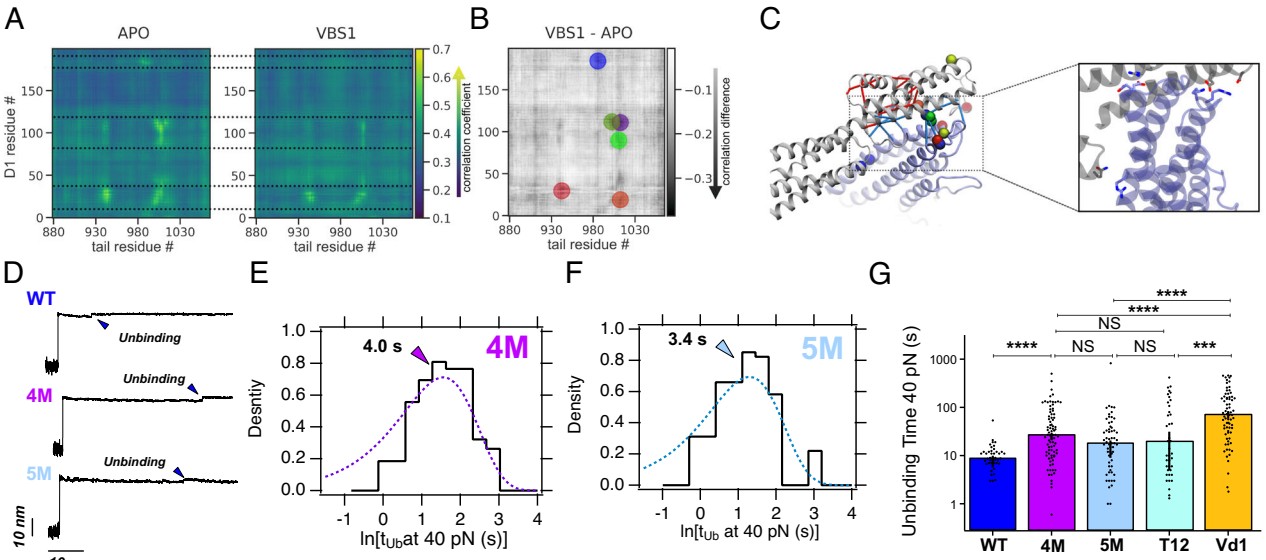

**Fig. 4 | Analysis of correlated motions and force distribution identified an overlapping set of residues that collectively weakens the D1-tail interface.** **A** The results of the analysis of correlated motions[79] in the tensed protein. (Extracted from force-probe MD trajectories at 0.1 m/s using the 5 ns leading up to a force of 300 pN across the protein in each trajectory. For each map, the contributions from 20 independent trajectories were averaged.) Here, residues of the vinculin tail are represented along the *x*-axis, and residues of the D1 domain along the *y*-axis. The areas between the dotted lines illustrate the three areas in the D1 domain that are strongly affected by VBS-binding. **B** Changes in correlation coefficient per residue between apo state and complex, such that dark areas represent residues that experience a strong loss of correlation if in complex with VBS1. The colored circles represent the results of a weighted cluster analysis, intended to find clusters of correlation-losing residues. **C** Representation of the results from the cluster analysis (colored spheres) and FDA-network analysis (red and blue connections). The zoom-in on the right shows the interactions between head and tail

residues (image based on the *x*-ray structure) that were identified by both methods to contribute to the lesser stability of the head–tail interactions. **D** Magnetic tweezers trajectories showing vinculin unbinding from talin R3[IVVI] at 40 pN for the WT (mature state, upper panel), the 4M mutant (middle panel), and the 5M mutant (lower panel). **E** Square-root histogram of unbinding times ($t_{Ub}$) at 40 pN for the 4M vinculin mutant. The single-peaked square-root histogram indicates a single-bound model. **F** Square-root histogram of unbinding times ($t_{Ub}$) at 40 pN for the 5M vinculin mutant. The 5M mutant also has a single-bound mode. **G** Unbinding times for WT vinculin (mature state, dark blue), 4 M (purple) and 5 M (light blue) mutant, the T12 mutant (teal), and D1 domain (orange). Data from $N = 42$ (WT mature); $N = 84$ (4 M); $N = 59$ (5 M); $N = 40$ (T12), and $N = 70$ (D1) vinculin unbinding events. Bars indicate average unbinding times, and error bars are SD. Significance levels for non-parametric Mann–Whitney test *NS P > 0.05*; ***$P < 10^{-3}$; ****$P < 10^{-4}$. Source data are provided as a Source Data file.

mechanism and its mimic by 4M and 5M are robust across different VBS, and to this end performed additional MD simulations with VBS3 and α-actinin (Fig. S8). VBS3 binding again caused a loss of correlations involving all those five residues, resembling 5 M, while alpha-actinin binding did not affect E986 interactions, resembling 4M (Fig. S8A). In accordance, VBS3 binding measurably weakened vinculin head–tail interactions, while alpha-actinin did not show significant changes within the simulated time scale (Fig. S8B). We predicted these mutants, due to the weakened head–tail interactions, to show stronger binding to talin, that is, binding times more comparable to the vinculin head domain than to full-length vinculin.

To test this predicted effect of the designed mutations (4M and 5M vinculin mutants) on the association between talin and vinculin, we used our magnetic tweezers assay to measure the stability of the mutant vinculin-talin interaction. When forcing the complex dissociation at 40 pN (Fig. 4D), we readily observed that the unbinding kinetics of the 4M and 5M mutants from talin were slower than those measured for WT vinculin, even compared to the mature state. Additionally, the unbinding times for both mutants were distributed as a single-exponential, indicating a single binding mode (Fig. 4E, F), which suggests that these mutants do not undergo any maturation transition upon binding to talin, at least within our experimental resolution. Finally, we probed the binding dynamics of the vinculin T12 mutant, which has previously been shown to exhibit a pre-opened conformation due to the weakening of the head-to-tail interaction[37]. T12 here shows similar properties as those of the 4M and 5M mutants (Fig. 4G and Fig. S9). Overall, these experiments place the binding affinity of the 4 M and 5M mutants between those

of the mature WT FLV and those of the D1 domain (Fig. 4G), which indicates that these designed mutations, as intended by our computational design, abolish the allosteric salt bridge network that directly allows forming a stronger complex with talin. These findings validate the allosteric switching mechanism at the D1-tail interface identified by the simulations and suggest that the mutations weaken the head–tail interface by mimicking the allosteric effect of VBS binding.

## Efficient bundling of actin filaments by the vinculin 5M and 4M mutants

We next examined the functional consequences of the designed vinculin mutants 4M and 5M in vitro. Vinculin has been shown to form stable actin bundles in the presence of activated talin[28,48]. Here, we used total internal reflection fluorescence (TIRF) microscopy to investigate the ability of the mutants to bundle actin filaments even in the absence of activated talin (Fig. 5). We monitored the polymerization of fluorescently labeled actin monomers in the presence of only the full-length vinculin variants or in the presence of variable amounts of the talin constitutively active vinculin-binding site 1 (VBS1, residues 482–636)[22,27,28] (Fig. 5A and movies S1 and S5). In the absence of talin-VBS1, the vinculin 5M variant induced stable actin bundles with a ratio of 40.3 ± 10.6 of the total population of filaments, 1.4 times greater bundles yielded by vinculin 4M variant (Fig. 5B). Surprisingly, the bundles formed in the presence of the vinculin 4M variant were less stable than those formed by the 5M variant (movies S2). Furthermore, in control assays we confirmed that the actin bundling in the absence of talin-VBS1 was due to the mutations in vinculin since the

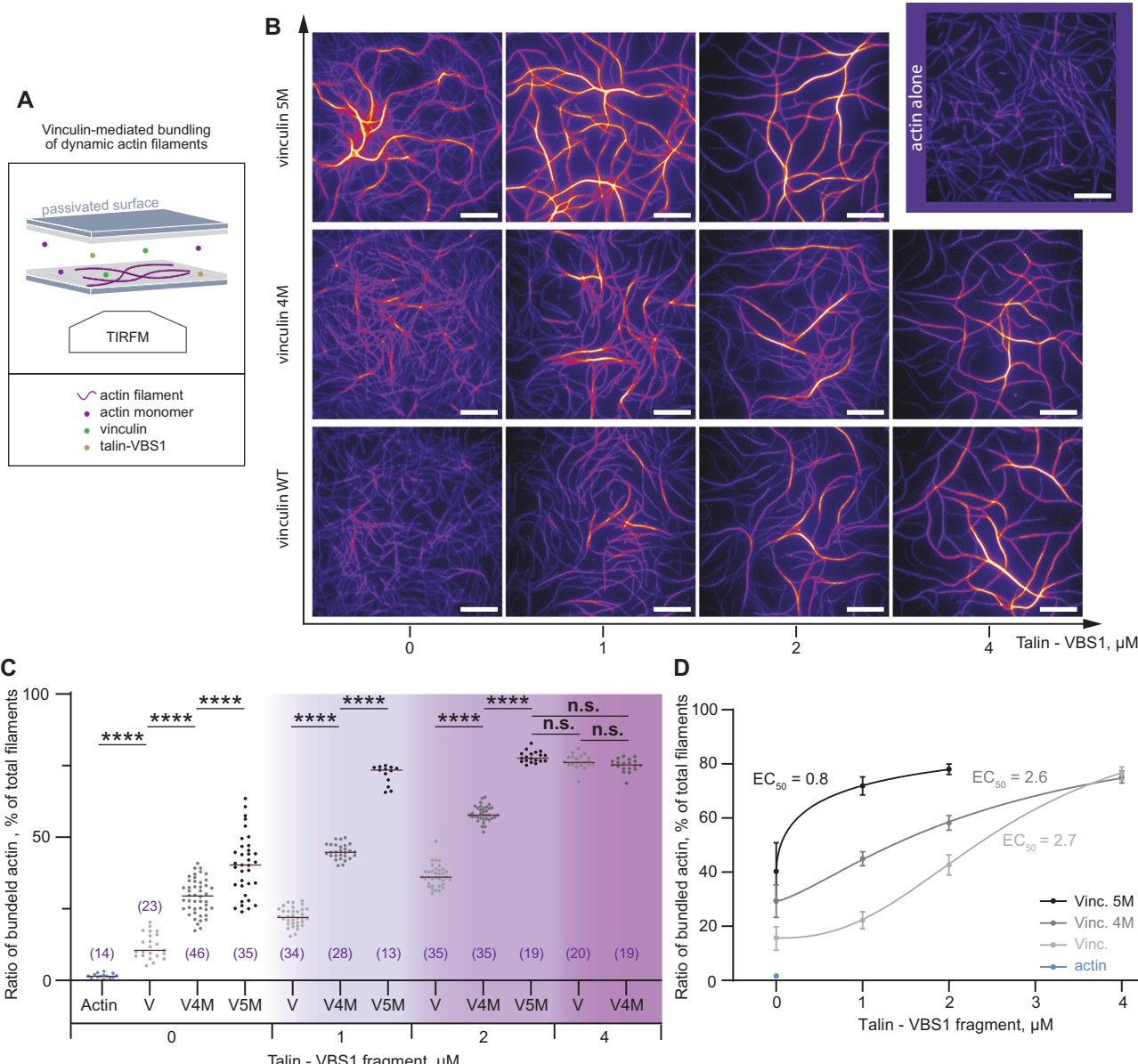

**Fig. 5 | The 5M vinculin mutant forms stable actin filament bundles in the absence of talin. A** Schematic illustration of the experimental setup. (1) The passivated slides and coverslips were utilized to fabricate a chamber where (2) soluble proteins were injected at the onset of polymerization/bundling reaction, as revealed by TIRFM. **B** Representative images after 1 h polymerization of 0.6 μM Alexa-647 labeled actin alone, or in the presence 0.5 μM vinculin variants, or in the presence of 0.35 μM vinculin variants and 1, 2, or 4 μM talin VBS1. Scale bars, 10 μm. **C** The relative amount of actin bundles was shown, as a ratio between bundled actin and the total filament population. Statistical comparisons using the Holm–Šídák test, and one-way analysis of variance (ANOVA) showed significant variations among vinculin variants for their ability to generate actin bundles. Number of actin bundles per condition shown in parentheses. Horizontal bar is the average value. ****$P < 10^{-4}$; NS.(V5M-VBS1 2 μM vs $V^{WT}$-VBS1 4 μM) $P = 0.9984$. NS.(V5M-VBS1 2 μM vs V4M-VBS1 4 μM) $P = 0.6842$. NS.($V^{WT}$-VBS1 4 μM vs V4M-VBS1 4 μM) $P = 0.9999$. Data from $N = 3$ independent reconstitutions. **D** Counts in (**C**) were plotted as a function of talin-VBS1 concentrations and fitted with a four-parameter dose–response equation, the Hill-Langmuir equation (see Methods). The best fit of the overall dataset was obtained for 99.8% maximal bundle ratio and showed a significant difference in the half-maximal effective concentration, EC50. $R^2$ ranged between 0.84 and 0.98, and was 0.95 for the overall dataset. Error bars represent SD. Source data are provided as a Source Data file.

polymerization kinetics of pure actin or supplemented with wild-type vinculin showed rarely and only momentarily actin filaments crossings (Fig. 5C and movies S1 and S2). This suggests that the mutations in the 4M protein, K944A, R945A, D1013A, E1015A, perturbed the head-to-tail interactions of vinculin and therefore induced actin bundling, while the additional mutation E986A significantly shifted vinculin dynamics towards a state prone to bind and stably bundle actin filaments (Fig. 5C). By adding increasing amounts of talin-VBS1 in the polymerization assays, we showed that the 5M variant has a significantly decreased half maximal effective concentration, EC50 (Fig. 5C, D, and

movies S3 to S5). Altogether these results suggest that while the K944A, R945A, D1013A, E1015A mutations yielded a slight increase in the affinity for actin filaments, the additional mutation at E986A position transforms vinculin into a more efficient actin bundler. The vinculin T12 mutant shows an overall similar behavior to that of 4M and 5M, but is less efficient than 5M in actin bundling in absence of VBS (Fig. S10). Since vinculin activation by VBS1 is the rate-limiting reaction for vinculin-mediated bundling, our joint simulation and in vitro data suggest that our mutations in vinculin mimicking VBS-binding render Vt more exposed and accessible for actin by diminishing the stabilizing

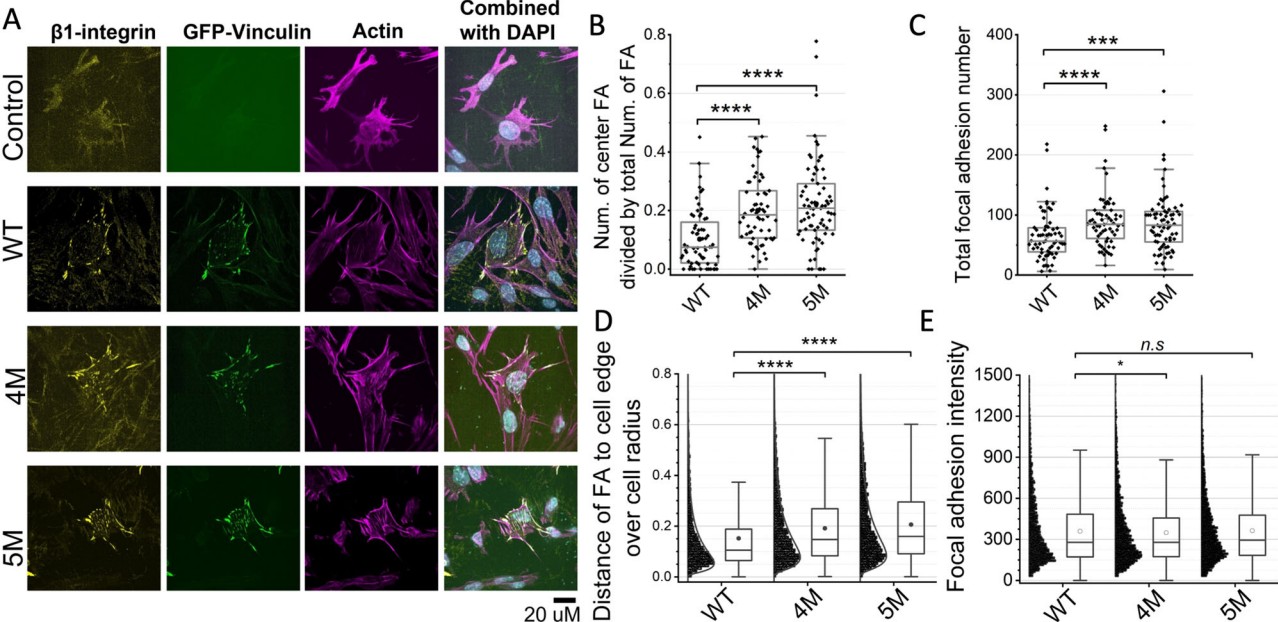

**Fig. 6 | Vinculin mutants mimicking VBS binding form centrally located adhesions distant from the cell edge. A** Images of MEF-vinculin-Null cells non-transfected, or transfected with GFP-wild-type vinculin, GFP-4M, or GFP-5M vinculin mutants. Cells were stained with $\beta_1$ integrin antibody, phalloidin and DAPI. Scale bar: 20 μm. **B** The fractions of center focal adhesion divided by the total number of focal adhesions of the cells. **C** Quantification of the total number of focal adhesions per cell. **D** Quantification of the shortest distance of each focal adhesion to cell edge normalized to the equivalent cell radius. **E** Quantification of focal adhesion intensity normalized to the background intensity. Box and whisker plot: The box includes the upper and lower quartile. The lower and upper whisker represents the lower quartile −1.5 *interquartile range and upper quartile +1.5 *interquartile range, respectively. The line in the box represents the median and the dots represent the mean. Significance levels from two-sided $t$ tests. NS (WT vs 4 M) $P = 0.06$; NS (WT vs 5 M) $P = 0.44$; **$P < 0.01$; ***$P < 0.001$; ****$P < 10^{-4}$. Source data are provided as a Source Data file.

salt-bridges at the Vt−Vh interface just as VBS binding allosterically does (compare Fig. 4B, C).

### Differential distribution of focal adhesions in vinculin-null MEFs following transfection with WT and mutant vinculin

Having established that the designed mutants mimic talin-activated vinculin in magnetic tweezers experiments and actin bundling assays, we next assessed the differential effects of WT and mutant (4M and 5M) vinculin forms on the formation and localization of cell-matrix adhesions in live cells. We transfected vinculin-null mouse embryonic fibroblasts (MEFs) with plasmids encoding the three molecules, tagged with GFP, to visualize the FA distribution in the transfected cells. Furthermore, the cells were fluorescently labeled for $\beta_1$ integrin and F-actin (Fig. 6A). Naturally, the non-transfected (Control) cells were GFP-negative, yet they displayed poorly organized $\beta_1$ integrin (detected by antibody labeling) and were lacking the typical morphology of FAs. It is noteworthy that the control, un-transfected cells did form FA-like structures that contain other adhesome components, such as zyxin, as shown in Fig. S11. Moreover, F-actin distribution in these cells was rather diffuse throughout the cytoplasm.

Expression of GFP-tagged WT vinculin induced a major reorganization of FAs and actin bundles, manifested by the formation of characteristic FAs, localized primarily at the cell periphery, and apparently recruiting b1 integrin into these sites. Expression of the mutant vinculin forms showed an enhanced capacity to form FAs, manifested mainly by the formation of multiple integrin adhesions that are located at the cell center.

To quantify these differences in FA formation, we segmented the vinculin adhesions in 62-75 representative cells (Table S2) and defined those that are located at the cell center and those located at the cells' periphery (for details, see Materials and Methods). As shown in Fig. 6B and D, the relative prominence of centrally located adhesions in the cells expressing the mutant vinculins (both 4M and 5M) was

significantly higher than in cells expressing WT vinculin. Furthermore, the average number of adhesions in cells expressing the mutant forms was ~1.4 larger compared to WT (Fig. 6C). Interestingly, the fluorescence intensity of FA was essentially the same in the three experimental groups (Fig. 6E).

To further explore the possible mechanism underlying the differential properties of "central" and "peripheral" FAs, we transfected the vinculin-null MEFs with WT and mutant vinculin forms that are not tagged by GFP, and then immunolabeled the cells for vinculin and zyxin, in addition to phalloidin and DAPI. The choice of zyxin, whose localization in FAs was shown to be force-dependent[38,49] was motivated by the assumption that peripheral FAs are exposed to higher shear stress than central adhesions (see e.g. ref. 50). The images presented in Fig. S11A further support the results obtained with the GFP-tagged WT and mutant vinculins, namely that WT vinculin supports the development of peripheral FAs only, while the mutant forms support also the formation of prominent central adhesions. Notably, zyxin showed an exclusive association with the peripheral adhesions (even in cells expressing the 4M and 5M mutants), and was virtually absent from the central ones, which suggest that the low shear adhesions in the cell center, that bind activated vinculin and not zyxin, might be fibrillar adhesions (see discussion section below). A quantification of the differential recruitment of the WT and mutant forms of vinculin is shown in Fig. S11B. Our observation supports the possibility that the mutations that were proposed to mimic the conformational changes that are physiologically driven by VBS1-mediated interaction of vinculin with talin indeed enable the mutated vinculin forms to bind and possibly support the lower-shear central FAs, which cannot recruit zyxin and WT vinculin.

### Discussion
Vinculin is a large multi-domain protein and a central hub of FAs and adherens-type cell−cell junctions that binds to the exposed vinculin-

binding sites (VBS) of talin and α-actinin. Force activates both talin and vinculin, and both proteins activate each other, but the molecular pathways of this force-dependent mutual allostery have remained poorly understood. We here lay out a comprehensive atomistic mechanism of the reciprocal talin-vinculin activation process, which we support by single-molecule experiments, actin-bundling assays, and live cell-based experiments using wild-type and prospectively constitutively activated mutant vinculin.

Using MD simulations, we identified here a network of key salt bridges at the head–tail interface of vinculin, which stall the protein in the auto-inhibited closed conformation. A rewiring of this network upon VBS binding then loosens the closed state and facilitates an unfolding of the molecule. While previous MD simulations on short time scales of several hundreds picoseconds could not reveal an effect of bound VBS on the force response of vinculin[30], our hundreds of nanosecond-long simulations resolve the activating effect of VBS at atomistic detail. Given the high structural and sequence conservation across different VBS, we propose this two-way allostery at the VBS-vinculin binding sites to be evolutionarily conserved and to also include non-talin VBS[19].

A large body of work has confirmed the need for talin to be activated for vinculin by VBS exposure (e.g[17,51,52]). A direct influence of force on vinculin activity has also been proposed but likewise questioned[31]. Our joint data underline that vinculin does not require to be subjected to force for binding talin. Instead, vinculin gets opened and activated upon talin binding in a largely force-independent manner through the intricate maturation mechanism we discovered. As a consequence, we find the designed vinculin mutant mimicking talin binding to localize to central adhesions in cells, which are known to be at lower shear[53] and to show fast dynamics according to fluorescence recovery after photobleaching experiments[54]. Previous studies (e.g. ref. [6].) demonstrated that cells spreading on fibronectin via α5β1 integrin develop focal adhesion throughout their entire ventral membrane (mostly "central adhesions"), while cells adhering to vitronectin via the αvβ3 receptors, form primarily "peripheral focal adhesions". Here, expression of wild-type vinculin rescues β1 integrin engagement in peripheral focal adhesions (along with zyxin), while expression of mutant vinculin in these cells leads to the association of β1 integrin in both peripheral and central adhesion sites. A similar observation has previously been made for the vinculin T12 mutant[55], in agreement with the comparable actin-bundling activity and behavior in the magnetic tweezers experiments observed for T12 and 4M/5M (Figs. S9, S10). These novel observations support the notion that vinculin plays a key role in the recruitment of β1 integrins (most likely α5β1) to adhesion sites (both peripheral and central) while zyxin association with focal adhesion is largely vinculin independent and highly force dependent.

Actomyosin-dependent forces acting on vinculin are not required for binding to talin but can, of course, further enhance the maturation by shifting the vinculin conformational ensemble further to the extended open state of vinculin[51,56–58]. In direct analogy, we predict the mechano-sensitive Focal Adhesion Kinase (FAK)[59,60], which requires force to detach the auto-inhibitory from the kinase domain to be active only in the peripheral but not the central FAs. Instead, its constitutively active mutant Y180A/M183A should be able to phosphorylate its downstream substrates even in the low-shear central FAs, enhancing FA assembly also in the center of cells.

Our single-molecule force spectroscopy experiments have provided us with quantitative insights into the dynamics of talin and vinculin binding, allowing us to propose, in conjunction with simulations, a mechanistic model on the force-dependent mutual regulation between talin and vinculin. Previous single-molecule studies using full-length vinculin binding to the talin R6 domain[36] revealed that full-length vinculin could bind talin, albeit with a lower probability than the vinculin head or the isolated D1 domain, in agreement with our observations on the R3 domain (Fig. 1 and Fig. S4). While these experiments monitored vinculin binding in an indirect way (and also, different from us, they used vinculin from bacterial expression, which lacks physiological post-translational modifications that might impact its binding properties), our capability to resolve individual binding and unbinding events in equilibrium over long timescales has allowed us to independently quantify the vinculin binding and unbinding kinetics, concluding that such decreased affinity arises from the much faster off-rates (Fig. 1F, G). This suggests that the D1 domain alone is responsible for recognizing and interacting with the stretched VBS. Vinculin binding to talin (on-rate) instead is ignorant about the vinculin activation state, different, for instance, than FAK, which requires activation to be able to fully interact with its partners.

More broadly, our results highlight the intricate role of mechanical forces in regulating the dynamics and interactions of mechanosensing proteins such as talin and vinculin. Mechanical forces control vinculin binding by requiring talin unfolding under force to expose the VBSs, and, upon binding, vinculin triggers a conformational change to reform the VBSs helices, precluding talin refolding. This interplay between talin nanomechanics and binding was also demonstrated with another key talin partner, deleted-in-cancer-1 (DLC1). DLC1 binds to the folded talin R8 domain, which suddenly gains an outstanding mechanical stability that makes it inextensible even under forces in excess of 100 pN[61], similar to the binding-induced mechanical stabilization of other non-force-sensing proteins[62,63]. However, different from the effect DLC1 binding has on talin mechanics, our study shifts the focus to the impact binding under force has on the binding protein. Vinculin undergoes a conformational change upon binding to talin that greatly increases the affinity of this interaction, but that can only occur over a tightly regulated force range, defined by the competition between vinculin activation and unbinding kinetics. At low forces (<10 pN), talin-vinculin binding is sufficiently long-lived to reach the mature state that greatly stabilizes this mechanosensing complex; however, if binding occurs under a higher tension (10-20 pN), vinculin quickly dissociates before reaching maturation, so the complex is very ephemeral. This mutual regulation between talin and vinculin establishes a much narrower force range (~5–10 pN) for successful vinculin binding than what previously thought, a range largely overlapping with the forces experienced by talin inside the cell (7–10 pN[9]). Interestingly, after maturation, forces on talin can increase up to ~20 pN without compromising vinculin binding as the active stable state has already been reached.

The rationally designed vinculin mutants 4M and 5M lack the maturation and instead functionally and constitutively mimic vinculin with VBS being bound. The results from actin bundling and cellular adhesion assays confirm the mutants to act as talin-bound vinculin molecules, with the 5M mutant showing even stronger actin bundling capacity than M4 or the previously described T12 mutant. The conservation of our mechanism across VBS observed in MD simulations suggests that the 4M and 5M vinculin mutants likely also serve as valuable mimics for other VBS-vinculin complexes such as those with alpha-actinin at cell–cell junctions. In fact, we observe VBS1 (and VBS3) to affect the additional mutation introduced in 5M more strongly than α-actinin, which could imply 5M to rather mimic the comparably strong binding of VBS1, while 4M rather captures vinculin binding to alpha-actinin.

In summary, our work identifies a vinculin allosteric mechanism that dynamically couples talin binding to vinculin activation and maturation of the complex. This allostery explains why the talin-vinculin interaction is kinetically stable, yet only at a very narrow force range of 7-10 pN, that is, in a regime where the formation of focal adhesions in the cell is desired. We propose 4M and 5M as versatile molecular tools to further dissect vinculin-mediated cellular mechanotransduction.

## Methods

### Molecular dynamics simulations−vinculin activation

The GROMACS[64] 2018.4 simulation package was used for all simulations discussed in this section. The AMBER-ff99sb-ILDN force field[65] with Joung ions[66], and the Tip3p water model[67] were utilized. Prior to production runs, the proteins were solvated, neutralized with a 0.15 M concentration of NaCl, and subjected to a 15 ps NVT and a 100 ps NPT equilibration. The NPT ensemble was kept at 300 K using the v-rescale thermostat[68] and pressure was controlled at 1 atm using the Parrinello-Rahman barostat[69] with a relaxation time of 2 ps. The use of hydrogen virtual sites[70] allowed for an integration time step of 5 fs. The 1TR2 full-length vinculin structure[23] excludes the proline-rich linker between residues P843 and P877. For our study, we inferred its conformation using the Chimera[71] interface to MODELLER[72–74].

To model the full-length VBS complexes, we used the pdb structures 1T01 (VBS1)[22], 1RKC (VBS3)[23], and 1YDI (ACT)[20] and combined them with the full-length vinculin structure 1TR2 adapting the procedure described previously[30]. Using Pymol, we generated a hybrid structure minimizing the RMSD between the Cα-atoms of the D1 domains. The resulting structures were energy minimized in vacuum after which the simulation box was filled with Tip3p water and 0.15 M NaCl, and the solvent energy was minimized by enforcing position restraints on the protein backbone and sidechains, which were gradually released in a three-step procedure in an NVT ensemble. Finally, an extensive 1 microsecond-long NPT equilibrium simulation was conducted to ensure proper relaxation of the complexes.

For the force-probe MD runs, force was applied to the center of mass of the tail domain, and on the talin binding site located on the D1 domain[23]. The apo-state recovery simulations were initiated from the resulting equilibrium conformation and, after removal of the VBS peptide, underwent the same energy minimization and equilibration scheme.

### Molecular dynamics simulations−peptide extension

Here, the GROMACS[64] version 2020.3 was used with the AMBER-ff99sb-ILDN force field with Joung ions[65,66] and the Tip3p water model. The simulation procedure was started with the VBS-D1 complexes structures with pdb-ID 1T01 (VBS1)[22]. Hydrogen virtual sites[70] allowed an increase of the integration time step to 5 fs in the production set of simulations. Temperature and pressure were controlled as described above. Prior to production runs, the complex of VBS and vinculin head was equilibrated for 100 ns. For the production runs, we used distance-pulling along the x-axis applying a constant force to the N-terminal and C-terminal C-α atoms of the respective vinculin binding site. For the VBS-vinculin-head complex, we simulated 20 × 1 μs-long runs for 4 different forces. In the case of the full-length vinculin complex, 20 × 50 ns were simulated for 5 different forces.

### Single-molecule magnetic tweezers experiments

The single-molecule magnetic tweezers experiments were conducted on a custom-made setup, as previously described[44]. Experiments are carried out in custom-made fluid chambers composed of two cover slides sandwiched between a laser-cut parafilm pattern. The bottom glasses are cleaned by sonication on a three-step protocol: (i) 1% Hellmanex at 50 °C, 30 min; (ii) acetone (30 min); (iii) ethanol (30 min), and then activated by plasma cleaning for 20 min to be silanized (immersion in 0.1% v/v 3-(aminopropyl)trimethoxysilane in ethanol for 30 min). Finally, the bottom cover slides are cured in the oven at 100 °C for >30 min. The top cover slides are cleaned on 1% Hellmanex at 50 °C, 30 min, and immersed in a repel silane (Cytiva, PlusOne Repel-Silane) for 30 min. The fluid chambers are assembled by melting the parafilm intercalator in a hot plate at 100 °C to ensure adhesion. After assembly, the chambers are incubated in a glutaraldehyde solution (glutaraldehyde grade I 70%, 0.001% v/v) for 1 h, followed by incubation with 0.002% w/v polystyrene beads (~2.5 μm diameter, Spherotech) for 20 min, and incubated with a 20 μg/ml solution of the HaloTag amine (O4) ligand (Promega) overnight. Finally, the chambers are passivated with a BSA-sulfhydryl blocked buffer (20 mM Tris-HCL pH 7.3, 150 mM NaCl, 2 mM MgCl2, 1% w/v sulfhydryl blocked BSA) for >3 h. The R3[IVVI] polyprotein construct is freshly diluted in PBS at ~2.5 nM and incubated for ~30 min to achieve HaloTag binding. Finally, streptavidin-coated superparamagnetic beads (Dynabeads M270, Invitrogen) are added to bind with the biotinylated polyprotein terminus for (~1 min). All experiments are performed on PBS buffer with 10 mM ascorbic acid (pH 7.3). After a talin molecule is found, full-length human vinculin is added to the fluid chamber by doing a buffer exchange to a buffer containing vinculin at the desired concentration and 10 mM ascorbic acid.

### Analysis of vinculin binding and unbinding kinetics

Individual binding events were detected as contraction or extension events in unfolded talin (see Fig. 1), allowing us to directly measure binding and unbinding times and, therefore, the binding and unbinding rates (Fig. 1F, G). The binding rates show a biphasic dependence with force, which first favors binding by unfolding talin and exposing the VBSs but then impairs binding due to the coil-to-helix transition triggered by vinculin binding, which is unfavored by high forces. We can model assuming that vinculin binding to an exposed vinculin binding site follows a simple Bell-Evans model[39,40], and that the probability of exposing a vinculin binding site follows the probability of talin unfolding, whose folding and unfolding rates can also be modeled by the Bell-Evans model. Therefore, the vinculin binding rates $r_B(F)$:

$$r_B(F) = k_B(F) * P_U(F) = k_0 e^{-\frac{Fx^\dagger}{kT}} \frac{k_0{}^U e^{F\frac{x^\dagger_U}{kT}}}{k_0{}^U e^{F\frac{x^\dagger_U}{kT}} + k_0{}^F e^{-F\frac{x^\dagger_F}{kT}}} \tag{1}$$

where $k_B(F) = k_0 e^{-Fx^\dagger/kT}$ are the vinculin binding rates to an exposed VBS and $P_U(F) = \frac{r_U(F)}{r_U(F) + r_F(F)}$ the unfolding probability of talin (exposing the VBSs), where $r_U(F) = k_0^U e^{Fx_U^\dagger/kT}$ and $r_F(F) = k_0^F e^{-Fx_F^\dagger/kT}$ are the unfolding and folding rates of talin, already characterized[41].

Similarly, the unbinding rates of vinculin follow Bell-Evans model:

$$r_{Ub}(F) = k_0^{Ub} e^{Fx_{Ub}^\dagger/kT} \tag{2}$$

Table S3 shows the fitting parameters obtained for the binding and unbinding rates of FLV and D1 to/from R3[IVVI].

### Square-root histograms

To identify the number of involved timescales in vinculin unbinding (Fig. 3C), we calculate the distribution of unbinding times with logarithmic binning. For an exponential distribution $p(t) = \frac{1}{t_0} e^{-t/t_0}$, being $t_O$ the characteristic timescale, a logarithmic binning is equivalent to applying the transformation $x = \ln t$. Therefore, the new distribution on $x$, calculated as $g(x) = p(t)\frac{dt}{dx}$, is $g(x) = \exp(x - x_0 - \exp(x - x_0))$, being $x_0 = \ln t_0$, which is the maximum of the distribution, hence easily identifiable as a peak. In this case we have two involved timescales (double exponential):

$$\begin{aligned} g(x) = &A^{(1)} \exp(x - x_0^{(1)} - \exp(x - x_0^{(1)})) \\ &+ A^{(2)} \exp(x - x_0^{(2)} - \exp(x - x_0^{(2)})) \end{aligned} \tag{3}$$

where $x_0^{(1)} = \ln t_0^{(1)}$, and $x_0^{(2)} = \ln t_0^{(2)}$ being $t_0^{(1)}$ and $t_0^{(2)}$ the two involved timescales (averages of each of the exponential distributions), identified as separate peaks in the distribution.

## Expression and purification of the magnetic tweezers polyproteins

The $(Ig32)_2$-$(R3^{IVVI})$-$(Ig32)_2$ polyprotein was engineered by restriction digestion using the compatible cohesive and restriction enzymes BamHI and BglII between BamHI and KpnI sites. pFN18a $(Ig32)_2$-$(R3^{IVVI})$-$(Ig32)_2$ was subcloned into a modified pFN18a vector engineered with the AviTag™ (Avidity) (sequence GLNDIFEAQKIEWHE). Constructs were cloned between the HaloTag at the N-terminal and the 6 histidine tag next to the AviTag™ at the C-terminal. Recombinant plasmids were transformed in XL1Blue (Agilent Technologies) or Top10 (Thermofisher scientific) competent cells.

Polyprotein constructs were expressed in E. Coli BLR(D3) cells (Novagen). Cells were grown in LB supplemented with 100 mg/ml ampicillin at 37 C. After reaching an $OD_{600}$ of 0.6, cultures were induced with 1 mM of IPTG and grown at 25 °C for 16 h. Cells were resuspended in 50 mM NaPi at pH 7.0 with 300 mM NaCl, 10% glycerol and 1 mM of DTT supplemented with 100 mg/ml lysozyme, 5 g/ml DNase, 5 g/ml RNase and 10 mM MgCl2 and incubated on ice for 30 min. This procedure was followed by gel filtration using a Superdex 200 10/300 GL column (GE Biosciens). Proteins were stored in gel filtration buffer 10 mM HEPES pH 7.2, 150 mN. NaCl, 10% glycerol and 1 mM EDTA at −80C.

Gel filtration fractions were pooled and concentrated using Amicron® filters with selected MWCO. Constructs were biotinylated using BirA Biotin Ligase (Avidity) following the manufactures suggested protocol. Biotinylation was confirmed using Streptavidin HRP conjugate (Millipore) with biotinylated/unbiotinulated MBP-AviTag™ fusion protein (Avidity) as controls.

## Human Vinculin mutagenesis for bacterial protein expression

Introduction of mutations into human vinculin (termed here M4 & M5) was based on the modeling described in Fig. 4A, B above. The mutations were introduced into a bacterial expression vector pBXNHG3_huVinculin (provided by Rajaa Boujemaa and Ohad Medalia, Univ. of Zurich) containing an N-terminal 10His tag and sfGFP-3C cleavage site. The mutations were introduced into the huVinculin coding sequence by the TPCR method[75] using Q5 High-Fidelity Master Mix (NEB). In the initial step the 4 mutations (K944A_R945A_D1013A_E1015A) were introduced, and the 5th (E986A) mutation was then added. The primers that were used to introduce the mutations are:

Vinculin_944A,945A_Forward:GAGGGGGCAGTGGTACCgctgcaGCACTCATTCAGTGTGCC

Vinculin_1013A,1015A_Reversed (used with the two forward primers): CTCTGTGGCCTGCTCAGAtgCCTCAgCACTGATGTTGGTCCGGC

Vinculin_986A_Forward: CAACCTCTTACAGGTATGTGccCGAATCCCAACCATAAGCACC

The entire ORF, including the promoter region was subsequently verified by DNA sequencing.

## Human full-length vinculin expression in insect cells

Full-length vinculin variants (wild type, V4M and V5M) cDNAs were amplified (using the forward 5′-ATATATGCTCTTCTAGTATGCCAGTGTTTC-3′, and the reverse 5′-TATATAGCTCTTCATGCCTGGTACCAGGG-3′ primers) of 1066 amino acids long, was cloned into a pFBXNH3 insect cell expression vector containing an N-terminal His tag followed by a 3C protease cleavage site, using the fragment exchange (FX) cloning strategy (Geertsma and Dutzler, 2011). Protein expression was performed by generating a recombinant baculovirus for ExpiSf9 insect cell infection at a density of $2.0 \times 10^6$ ml$^{-1}$ using the Bac-to-Bac system (Invitrogen). Briefly, the recombinant vinculin encoding plasmid pFBXNH3 was transformed into DH10Bac E. coli cells, which enabled the transposition of the recombinant gene into the bacmid genome. The recombinant bacmid DNA was then isolated and transfected into ExpiSf9 cells to generate the full-length vinculin-expressing baculovirus.

## Protein expression, purification and fluorescence labeling

ExpiSf9 cells were infected with the full-length vinculin-expressing baculovirus for recombinant protein expression as described[28]. Vinculin T12 mutant was expressed as described[37]. Briefly, expressing cells were harvested by gentle centrifugation and lysed by sonication in 20 mM Tris-HCl pH 7.5, 0.4 M NaCl, 5% Glycerol, 1 mM DTT, 0.1% Triton, and protease inhibitors. Full-length vinculin variants were purified from clarified cell extract using a Ni-Sepharose 6 Fast Flow metal affinity chromatography, HisTrapFF (Cytiva Lifesciences). Proteins were further purified on a Superdex 200 size exclusion column (Cytiva Lifesciences) and eluted with 20 mM Tris-HCl pH 7.8, 0.15 M KCl, 1 mM MgCl2 and 5 % Glycerol. Protein aliquots were supplemented with 50% glycerol and stored at −20 °C.

Actin was purified from rabbit skeletal muscle acetone powder, purified according to the method of Spudich and Watt[76], and stored in G-buffer (5 mM Tris·HCl pH 7.8, 0.2 mM CaCl2, 0.5 mM DTT, 0.2 mM ATP). Actin was further labeled on cysteine with Alexa-Fluor 647 maleimide dyes (Invitrogen). Labeling was performed in 2 mM Tris HCl pH 7, 0.2 mM $CaCl_2$, 0.2 mM ATP, for 16 h. To limit excessive labeling, protein/dye molar ratio was kept ≤ 1:3. The reactions were stopped by adding 10 mM DTT. Labeled actin was polymerized, depolymerized, and gel-filtered using Superdex 200 size exclusion column (Cytiva Lifesciences), in 2 mM Tris HCl pH 7, 0.2 mM $CaCl_2$, 0.2 mM ATP, 0.5 mM DTT. Ca-monomeric actin was stored on ice and used within 2 weeks for TIRFM. Human Talin 1 VBS1 residues 482-636 construct was expressed in BL21 (DE3)pLysS strain and purified using using a Ni-Sepharose 6 Fast Flow metal affinity chromatography as described in ref. 28, supplemented with 20% glycerol, aliquoted, flash-frozen in liquid nitrogen and stored at −80 °C.

## Glass surface passivation

Slides and coverslips (CVs) used to assemble the reaction chambers were drastically cleaned by successive chemical treatments, and coated with tri-ethoxy-silane-PEG (5 kDa, PLS-2011, Creative PEGWorks, USA) 1 mg/ml in ethanol 96% and 0.02% of HCl, as described previously[28]. mPEG-silane coated slides and CVs were then stored in a clean container and used within a week time.

## TIRFM imaging

Reconstitution assays were performed using fresh actin polymerization buffer, containing 20 mM Hepes pH 7.0, 40 mM KCl, 1 mM MgCl2, 1 mM EGTA, 100 mM β-mercaptoethanol, 1.2 mM ATP, 20 mM glucose, 40 µg/mL catalase, 100 µg/mL glucose oxidase, and 0.4% methylcellulose. The final actin concentrations were 0.6 µM, with 20% Alexa monomer labeled. The polymerization medium was supplemented with vinculin variants and talin VBS1, as indicated in the figure legends and methods. The reaction medium was rapidly injected into a passivated flow cell at the onset of actin assembly, imaging started after 2 min. Time-lapse TIRFM was recorded every 15 sec. TIRF images were acquired using a Widefield/TIRF Leica SR GSD 3D or Leica TIRF microscope. The Widefield/TIRF Leica SR GSD 3D microscope was an inverted widefield microscope (Leica DMI6000B/AM TIRF MC) equipped with a ×160 objective (HCX PL APO for GSD/TIRF, NA 1.43), a Leica SuMo Stage, a PIFOC piezo nanofocusing system (Physik Instrumente, Germany) to minimize the drift for an accurate imaging, and combined with an Andor iXon Ultra 897 EMCCD camera (Andor, Oxford Instruments). The Leica TIRF microscope was an inverted widefield microscope (Leica DMI6000B) equipped with a ×100 objective (HCX PL APO for TIRF, NA 1.47), a motorized X-Y Stage, an Adaptive Focus Control (AFC) to correct for the Z drift, and combined with an Andor iXon EMCCD camera (Andor, Oxford Instruments). Fluorescent proteins were excited using a solid-state diode laser

642 nm (500 mW). Laser power was set to 3% and dyes were excited for 50 ms. Image acquisition was performed with 25 degrees-equilibrated samples and microscope stage. The microscope and devices were driven by Leica LAS X software (Leica Microsystems, GmbH, Germany).

### Image processing and data analysis of fluorescence images

Time-lapse videos of filament growth and steady-state images taken after 1 h of actin assembly were processed with Fiji software (NIH). 15 to 35 steady-state images were taken for each condition. To determine the ratio of bundled actin to the total assembled filaments, macros written in Fiji allowed to first subtract the background for each individual image, then average the min and max fluorescence intensities for single actin filaments, bin gray values (256 per 65535 gray levels) and calculate histogram counts for individual images. This allowed us to evaluate counts of pixels occupied by single actin filaments or by bundled actin. Data (Figs. 5d and S10) was fitted with a dose–response curve using GraphPad ("[Agonist] vs. response, Variable slope"). The equation was

$$Y = b + \frac{\left(X^{HillSlope}\right)(a - b)}{\left(X^{HillSlope} + EC_{50}^{HillSlope}\right)}, \tag{4}$$

where $a$ and $b$ are plateaus in the units of the $Y$-xis, $HillSlope$ describes the steepness of the curve, $EC_{50}$ is the concentration that gives a response halfway between $a$ and $b$. Three parameters, $EC_{50}$, $HillSlope$, and $b$, were unconstrained. The best-fit values for the four parameters, plateaus, Hill slope and EC50 were calculated from the overall dataset, with the value of the saturation plateau to be shared between all datasets.

### Human Vinculin mutagenesis for mammalian expression

The bacterial Vinculin mutant constructs were served as templates for PCR amplification of 10His-GFP-3C-Vinculin K944A_R945A_D1013A_E1015A (huVin_M4) or K944A_R945A_D1013A_E1015A E986A (huVin_M5) mutant sequences. The resulting PCR fragments were cloned into pCDNA3.4 by restriction enzymes and ligation. Using different sets of primers but the same bacterial vinculin mutant constructs as templates. 10His-Vinculin mutant sequences, without GFP-3C sequences were PCR amplified and cloned into pCDNA3.4 by restriction enzymes and ligation. The WT Vinculin sequence was cloned similarly by restriction enzyme and ligation to obtain 10His-Vinculinwt_pCDNA3.4 construct.

### Cell culture

Vinculin-null mouse embryonic fibroblasts (Vin-null-MEFs), derived from vinculin-null mice, were generously provided by Eileen Adamson (Burnham Institute)[77]. The cells were grown in DMEM (Gibco, Grand Island, New York) containing 10% FCS and 100 U/mL PenStrep (Biological Industries, Beit Haemek, Israel) at 37 °C in a 5% CO2 humidified atmosphere.

### Expression of GFP-tagged vinculin variants (WT and 4M/5M mutants) in Vin-null MEFs

The vinculin-null MEFs were seeded onto optical glass-bottom 96-well plates (Cat#164588 Thermo Scientific Nunc) coated with 10 μg/ml Bovine fibronectin (Biological Industries –Currently Sartorius). After 24 h, the cells were transfected with the different vinculin variants, tagged with GFP, using TurboFect (Thermo Fisher Scientific, Waltham, MA) as a transfection agent. The transfection was conducted according to the manufacturers' protocols. After 24 h, the cells were simultaneously permeabilized and fixed, with 0.5% Triton X100 in 3% paraformaldehyde for 2 min and further fixed with 3% paraformaldehyde for additional 30 min. Then, the cells were stained with beta-1 integrin antibody (HUTS21, BD Bio-sciences) or mouse anti-human β1 integrin

P5D2 (Developmental studies Hybridoma bank)", Phalloidin TRITC (Sigma) and DAPI (Sigma).

### Expression of His-tagged vinculin variants in vinculin-null MEFs

His-tagged vinculin variants (WT and the 4M/5M mutants) were transfected into Vin-null- MEFs, and the cells were processed as above and co-stained for vinculin (monoclonal) and zyxin (polyclonal, both prepared by the Antibody Production Laboratory of the Department of Biological Services, Weizmann Institute of Science), Phalloidin TRITC (Sigma) and DAPI (Sigma).

### Imaging

Fluorescence images were acquired using a DeltaVision Elite system (GE Healthcare/ Applied Precision, USA), mounted on an inverted IX71 microscope (Olympus, Japan) and equipped with a Photometrics CoolSNAP HQ2 camera (Photometrics, Tucson, AZ). The system was running SoftWorX 6.1.3. Pictures were acquired using an Olympus UIS2 BFP1 60×1.42 PlanApoN oil objective (Olympus, Japan).

### Image analysis

Confocal images or wide-field fluorescent images of vinculin or zyxin were first segmented using Ilastik software (https://www.ilastik.org). After the segmentation, individual objects were identified, and the object with a size larger than 0.3 μm² were preserved and considered as a "focal adhesion-like structure". Confocal or wide-field fluorescent images of actin were used to identify cell boundaries. Intensity thresholding by Otsu's method[78] was applied to segment the cells. After the segmentation, the object with the largest area was preserved and considered as the main cell body. The approximate cell radius was then defined as

$$r = \sqrt{\frac{area}{\pi}}. \tag{5}$$

Quantification of FA area, length and fluorescence intensity was performed using a custom code written in Matlab (MathWorks, Natick, MA) after correcting for fluorescence bleaching.

### Focal adhesion intensity measurement

Since the intensity of focal adhesions and related structures can be influenced by the background intensity of the cell, especially in wide-field fluorescent images, we normalized the focal adhesion intensity by subtracting the background intensity. The local background intensity was calculated for each focal adhesion by measuring the average intensity of the area surrounding the focal adhesion. The area outside of the cell and other focal adhesions are excluded from the background intensity calculation. Average intensity of each focal adhesion area was then calculated after the subtraction of background intensity.

### Focal adhesion location measurement

The distance between a focal adhesion and the cell's edge was obtained by calculating the shortest distance between the center of mass of the focal adhesion and the cell boundary. The "central focal adhesions (possibly—"fibrillar adhesions") were defined as those adhesions whose distance from the cell edge is larger than $0.3 \times r$(*approximate cell radius*).

### Reporting summary

Further information on research design is available in the Nature Portfolio Reporting Summary linked to this article.

## Data availability

The data generated in this study are provided in the Source Data files. All Source Data files are publicly available: https://doi.org/10.11588/data/WZDKT6.

## Code availability

All code is available upon reasonable request to the corresponding authors. Code will be made available and assistance provided as soon as possible but always within four weeks after the request.

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

## Acknowledgements

F.G. and B.G. thank the Klaus Tschira Foundation for generous financial support. O.M. and R.B.P. thank the ZMB and ScopeM microscopy centers at UZH and ETH, respectively, for their support and assistance. The study was also supported by the Minerva Center on "Aging, from Physical Materials to Human Tissues" to B.G.; the Schweizerischer Nationalfonds zur Förderung der Wissenschaftlichen Forschung to O.M. (SNSF 310030_207453); the HGS MathComp graduate school of Heidelberg University, the Max Planck School Matter to Life supported by the German Federal Ministry of Education and Research (BMBF) in collaboration with the Max Planck Society, the Flagship Initiative funded by the Federal Ministry of Education and Research (BMBF) and the Ministry of Science Baden-Württemberg within the framework of the Excellence Strategy of the Federal and State Governments of Germany, the state of Baden-Württemberg through bwHPC and the German Research Foundation (DFG) through grant INST 35/1134-1 FUGG to F.G. This work was supported in part by the Francis Crick Institute which receives its core funding from Cancer Research U.K. (FC001002), the U.K. Medical Research Council (FC001002), and the Wellcome Trust (FC001002). R.T.-R. is recipient of a King's Prize Fellowship. This work was supported by the European Commission (Mechanocontrol, Grant Agreement 731957), BBSRC sLoLa (BB/V003518/1), Leverhulme Trust Research Leadership Award RL 2016-015, Wellcome Trust Investigator Award 212218/Z/18/Z and Royal Society Wolfson Fellowship RSWF/R3/183006 to S.G.M, and by the European Commission (RADICOL, 101002812) to F.G.

## Author contributions

F.F. and F.G. conceived the research. F.F. performed the molecular dynamics simulations, and F.F. and C.A.S. analyzed the data. R.T.R. and S.G.M. conceived the single molecule experiments and RTR conducted and analyzed them. O.M. and R.B.P. purified all the proteins except the single molecule constructs and performed the actin-bundling assays. B.G., S.W.K., and W.L. contributed to cell-based experiments and analysis. F.F., T.U., and S.A. designed the mutant vinculin and T.U. and S.A. prepared the mutant vinculin constructs. All authors contributed to data analysis and wrote the manuscript.

## Funding

## Competing interests

The authors declare no competing interests.
