## [Peer review file · Nature Communications]

REVIEWER COMMENTS

Reviewer #1 (Remarks to the Author):

In this paper the authors combine the manipulation of single molecules of talin and vinculin by magnetic tweezers, molecular dynamics simulations, actin bundling assays and adhesion assembly experiments in cells to show that vinculin undergoes a maturation process upon talin binding which reinforces its binding to talin.

Comments:

The data in Figure 1D-H are generated entirely with a domain of talin artificially stabilized by mutations reported in a paper from another group (talin R3 IVVI). The authors should justify the use of this construct rather than the WT. The authors claim that vinculin affinity for the VBS contained in this domain is not altered. Since the force applied to this domain favors vinculin binding and this domain is more stable than the WT, I conclude that the mutation necessarily affects vinculin binding, at least indirectly. It would have been better to compare the data with the WT or to use only the WT. Here one can question the relevance for the cell since the conclusions are obtained with a construct that has a much higher mechanical stability than talin in the cell.

In Figure 3C, the authors cannot fit their distribution with a double exponential. This distribution is not precise enough. It shows a single highly populated class for short times, while the rest of the classes for other times are sparsely populated.

Figure 3D is a convincing demonstration of the maturation mechanism described by the authors.

The 5M mutant appears to be open, i.e. the head and tail are dissociated. If the head is exposed in this mutant, I do not understand what more Figure 4D-F tells us about maturation than what is shown in Figure 3E, which shows that the vinculin head no longer undergoes maturation because it binds constitutively stably.

The observation of larger focal adhesions in the center of cells expressing vinculin mutants does not seem to me to demonstrate the existence of a maturation phenomenon of vinculin that would strengthen its link with talin over time. I suggest to the authors that an interesting experiment would have been to measure the dissociation rate of WT vinculin by FRAP in increasingly older focal adhesions (from the leading edge to the center), i.e. containing talin-vinculin complexes with increasingly longer lifetimes.

The novelty of the conclusions derived from figures 1, 2 and 3 is limited as indicated by the titles of the figures. Fig.1 : Vinculin's head-tail interface regulates vinculin-VBS binding ; Fig.2 : Talin binding facilitates vinculin head-tail opening ; Fig :3. Talin binding weakens the vinculin head-to-tail interaction. These conclusions confirm what is already documented and well established in the field by a large amount of publications that are unfortunately not all cited here. The methodology which consists in putting under tension single molecules of talin and vinculin by magnetic tweezers has also been used in the past on several occasions by several laboratories. Moreover, this setup has been used by "Wang et al. Force-Dependent Interactions between Talin and Full-Length Vinculin. J Am Chem Soc. 2021" to determine that mechanically exposed VBS binds vinculin significantly more tightly than the isolated non-stretched VBS alone, which is also the conclusion of the present study by Franz et al. However, I agree that the concept of a vinculin maturation that reinforces its link with talin after binding is a new and interesting discovery.

The 5M mutant of vinculin is novel and will likely be useful in mimicking the active conformation of vinculin. However there are already several active vinculin mutants such as the most widely used called T12. The bundling activity of vinculin has also been reported several times and it is already accepted that this activity is carried by Vt exposed following vinculin activation. In the context of the present study, it is not clear what the use of this mutant in cells or in bundling assays add to the demonstration of a maturation process of vinculin. Other studies have shown that vinculin activation promotes focal adhesion formation and actin binding.

In the discussion section, many references should be cited for the following sentence "Actomyosin-dependent forces, acting on vinculin are not required but can, of course, further enhance the maturation by shifting the vinculin conformational ensemble further to the extended open state of vinculin."

Although the technical quality of this study is excellent and the discovery of vinculin maturation is interesting, it must be said that the novelty of the concepts presented is somewhat limited for a wide audience.

Reviewer #2 (Remarks to the Author):

In the manuscript "How talin allosterically activates vinculin" the authors describe a maturation mechanism between vinculin and force-activated talin. Full length vinculin forms a less stable complex with talin than the isolated (vinculin-binding) D1 domain, as the authors convincingly show that the head-tail internal interaction of vinculin weakens its external interaction with vinculin binding sites (VBS) fragments through an allosteric mechanism. This finding supports a mechanism where a more stable complex between VBS and vinculin requires its head to be engaged in a competing interaction with actin. Interestingly, the authors find that a maturation mechanism taking place after a few seconds and at intermediate forces, can strengthen the interaction of vinculin with the VBS from talin. Equally important, full-length vinculin, which is known to be in an autoinhibited form, seem to be able to bind to exposed vinculin binding sites (VBS). The authors then use molecular dynamics simulations to describe the mechanism in detail and find two mutants that would enhance the vinculin-VSB interaction.

Major comments:

- 1. In figure 1 a comparison is made between the response to force of the VBS fragment in the presence of vinculin head and full length vinculin. The authors seem to be missing a critical control. What happens when the VBS fragment is pulled without vinculin being present? They should add those passage times to Figure 1c.**
- 2. Do the two Ig32 domains unfold at higher forces during the magnetic tweezers experiments, when exposed to higher forces? Could they constitute a fingerprint for this molecular system?**
- 3. The authors state that their setup with the magnetic tape operates in the 0-40 pN range. However, the data in figure 1G shows unbinding rates at 60 pN. Are those rates coming from the magnetic tweezers experiments or from simulations? If not, clarify this discrepancy between the working range of the magnetic tweezers and the applied force in Fig. 1G.**
- 4. The fast part of the double exponential relies on fitting a single bin, which probably does**

not produce a statistically significant value, as is dominated by the size of the bin used to make the histogram. Maybe a histogram of the logarithm values of the measured unbinding times can produce more accurate values on the fast and slow unbinding rates.

5. The dual behavior for unbinding of vinculin when attached to talin is a very interesting result. The authors find that the vinculin domain binding talin (Vd1) and full-length vinculin (WT) have similar unbinding kinetics from talin R3 domain, supporting the idea that this dual behavior is not related to the direct interaction between talin and vinculin. The authors propose that vinculin can undergo a slow conformational change when bound to a VBS of talin and even propose two mutants and a molecular mechanism for it. However, in their simulations the VBS fragment does not seem to be under force. It was recently shown that mechanical activation of talin R8 can induce mechanically stable states when ligands bind. Is this the case for the VBS site? Simulations with this fragment under a force vector could give some insight.

6. In a different article, also not cited here, vinculin-VSB interaction was shown to have a reinforcement mechanism occurring along preferential force directions (10.1016/j.bpj.2019.12.042). This article seem to also propose a time and force dependent binding mechanism. Can the authors comment on how their findings are similar/different?

7. The model proposed here (Figure 3F) claims that the stable state is a force activated state that requires a low force (~ 8 pN) and would not exist under a high force (~ 15 pN). Do the MD simulations measured both of these states in a force-dependent manner?

8. Furthermore, to confirm the hypothesis mentioned at the points above, the authors performed single molecule measurements of talin unbinding in the presence of vinculin mutants and see a significant change in the unbinding time of these mutant when compared to WT vinculin (Figure 4F). The times reported in this figure for WT (~ 10 s) and Vd1 (~ 100 s) seem to however disagree with the statement that the authors make, that the unbinding kinetics of Vd1 and WT are similar (first and last bar in Figure 4F). Please explain this difference.

9. Also the authors claim that the 4M mutant does not undergo any maturation process. However the rate measured for unbinding of this mutant (Figure 4E – showing 4M mutant that does not undergo the maturation process) is much slower than the maturation rate for the WT vinculin (Figure 3C). These results suggests to me that the mutant rather undergoes maturation at much faster rate and the measured state is rather related to this 'mature' state.

10. The microscopy based data seems to contradict the single molecule measurements, with one showing that mutant 4 M forms a more stable complex than 5M, while the other the opposite. How do the authors reconcile these results?

11. What is the scale of the images in Figure 6A? Also the font size used in Figure 6B-E is too small to be easily read.

12. Several single molecule studies have shown so far a binding-induced stabilization effect, both on the proteins reported here, as well as other proteins. A small discussion on this topic and comparison with those studies would greatly benefit this manuscript.

13. The magnetic tweezers data is of very high quality and provides the strongest support for the conclusions of the paper. In the main text Figure 1, 3 and 4 rely on these data. According to author contributions, those measurements were performed by the second author, who in my opinion deserves at least equal contributor/co-first author status, given

the amount of work he put in compared to the other measurements reported in this manuscript.

Minor comments:

According to Merriam Webster dictionary, the noun "tweezers" is "plural in form but singular or plural in construction". Correct the abstract where it appears as "magnetic tweezer" to read "magnetic tweezers", which is also the correct name for this single molecule technique. On the other hand, also in the abstract, "two-ways network" should be "two-way network".

The authors could probably use the same 20 s scale bar in both panels of Figure 3.

Page 1 – the part of the second sentence starting with "when the tiny ..." seems to be missing the verb

Page 2 – remove the comma between "10" and "pN"

Page 2 – replace "composed of 1066-residue-long" to "composed of 1066 residues". Also check that indeed vinculin has 1066 amino acids. Uniport.org reports 1134 amino acids for human vinculin.

Page 4 – clarify what you mean by "actin is not able to bind to on its own"

Page 5 – under "Results" second and third sentences need some rephrasing or are missing words

Reviewer #3 (Remarks to the Author):

The paper by Gräter et al. presents an important combined experimental and MD simulation work resulting from the cooperation of several groups. The activation mechanism of vinculin and interaction with talin is studied, and fundamental results for actin bundling and cell association are obtained.

Unfortunately the quality of presentation is by no way adequate to the reported results, please regard the comments I made to the original manuscript in the appended file:

- The text contains very confusing unclear phrases.
- Important details on numerical analysis (Bell-Evans, dose-response) and simulation have not been given.
- The important results are obscured by poor writing, the need of presenting many details and the lack of a conclusion section.

The resulting manuscript is neither an instructive short communication of important findings nor a satisfying self contained report on their work.

I suggest that the authors first publish a letter focusing on the most important conclusions and then a really complete report elsewhere.

Reviewer #1 (Remarks to the Author):

In this paper the authors combine the manipulation of single molecules of talin and vinculin by magnetic tweezers, molecular dynamics simulations, actin bundling assays and adhesion assembly experiments in cells to show that vinculin undergoes a maturation process upon talin binding which reinforces its binding to talin.

Comments:

The data in Figure 1D-H are generated entirely with a domain of talin artificially stabilized by mutations reported in a paper from another group (talin R3 IVVI). The authors should justify the use of this construct rather than the WT. The authors claim that vinculin affinity for the VBS contained in this domain is not altered. Since the force applied to this domain favors vinculin binding and this domain is more stable than the WT, I conclude that the mutation necessarily affects vinculin binding, at least indirectly. It would have been better to compare the data with the WT or to use only the WT. Here one can question the relevance for the cell since the conclusions are obtained with a construct that has a much higher mechanical stability than talin in the cell.

The reviewer raises a good point, which we considered in the design of the experiments. The R3IVVI domain is a convenient tool for investigating vinculin binding due to its higher mechanical stability and slower folding kinetics, which amplify the signature for vinculin binding, allowing for a more thorough quantification. The mechanically weaker R3WT exhibits faster and noisier folding/unfolding dynamics, which makes it difficult to identify binding events at low forces, in particular between ~7-10 pN where the binding contraction is too small to be resolved within the molecular noise. Previously, we demonstrated that the binding mechanism of the vinculin Vd1 domain was irrespective of the talin R3 variant (WT or IVVI); simply, the lower mechanical stability of R3WT (~5 pN unfolding force vs ~9 pN for R3IVVI) lowers the threshold force for vinculin binding (*Tapia-Rojo et al. Sci Adv 2020*). In this sense, the affinity of vinculin for unfolded R3WT or R3IVVI is the same and, as the reviewer suggests, it only affects binding in an indirect way, due to the different unfolding force of each variant.

Following the reviewer's suggestion, we have now conducted new magnetic tweezers experiments to investigate binding of full-length vinculin to talin R3WT, and reached the same findings as for R3IVVI. These results are shown in new Figs. S4 and S6. First, full-length vinculin binding to R3WT or R3IVVI is alike, simply showing a lower threshold force for binding that correlates with the unfolding forces of R3WT (Fig. S4). Moreover, full-length vinculin similarly matures upon binding to R3WT, evolving to form a tighter molecular complex (Fig. S6). The only difference we found is that the maturation kinetics seem faster when binding to R3WT (~11 s vs ~37 s for R3IVVI). This could arise from the lower forces employed here (5 pN, the coexistence force for R3WT) that could allow for a faster maturation. Overall, these new data demonstrate that the binding mechanism and

dynamics of full-length vinculin are overall irrespective of the R3 variant to which it binds.

With respect to the cell studies, it was previously demonstrated that mouse fibroblast expressing talin containing the IVVI mutation on the R3 domain showed a higher rigidity threshold (from ~7 kPa to ~12 kPa) for vinculin binding and YAP translocation to the nucleus (*Elosegui-Artola Nat Cell Bio 2016*). In our experiments, the cells were plated on a stiff substrate (glass), well above the rigidity threshold; therefore, it can be assumed that the mechanical stability of the talin R3 domain plays no role.

In Figure 3C, the authors cannot fit their distribution with a double exponential. This distribution is not precise enough. It shows a single highly populated class for short times, while the rest of the classes for other times are sparsely populated.

We thank the reviewer for raising this point. We have now calculated and plotted the dwell-time distribution with logarithmic binning (Fig. 3C). With this conversion (often dubbed square-root histogram), an exponential distribution takes the following form: $p(x) = A \cdot \exp(x - x_0) \cdot \exp(-\exp(x - x_0))$, being $x = \ln t$ and $x_0 = \ln t_0$, and x_0 appears now as a peak in the distribution. Therefore, a double-exponential distribution shows two peaks, allowing for an easier identification of the involved timescales in a kinetic process. Square-root histograms are commonly employed to analyze ion channel recordings (*Sigworth Biophys J 1987*), were used to unveil complex enzymatic kinetics in single-molecule data (*Szozkiewicz Langmuir 2008*) and we previously used them for detecting molten globule-like states in the folding pathway of protein L using magnetic tweezers (*Tapia-Rojo PNAS 2019*). The square-root histograms of the unbinding times of full-length vinculin for R3IVVI (Fig. 3C) and R3WT (Fig. S4) show two peaks, indicating two underpinning characteristic times (corresponding to the weak and mature-bound state). Conversely, using the same data-analysis approach, the square-root histograms for the unbinding times of the 4M, 5M (Fig. 4E,F), and T12 (Fig. S9) mutants, and vinculin head (Fig. S7), show a single peak indicating one involved timescale, indicative of a single bound mode.

Figure 3D is a convincing demonstration of the maturation mechanism described by the authors.

Thank you for this positive response.

The 5M mutant appears to be open, i.e. the head and tail are dissociated. If the head is exposed in this mutant, I do not understand what more Figure 4D-F tells us about maturation than what is shown in Figure 3E, which shows that the vinculin head no longer undergoes maturation because it binds constitutively stably.

The results shown in Figure 4D-F (now 4D to 4F) are magnetic tweezers experiments to characterize the unbinding kinetics of the 4M and 5M mutants; in particular, Figs. 4E and 4F indicate that these mutants have a single bound-mode, undergoing no maturation, at

least within our time resolution. Their additional value is the confirmation of the allosteric switch observed in the MD simulations (Figure 4A-C). Since the maturation of WT vinculin is related to its activation (opening) and given that 4M and 5M mutants do not mature (directly showing a strong interaction with talin) these single-molecule experiments imply that the maturation can be attributed to an allosteric adaptation of some key D1-tail interactions resulting in the weakening of the interface. We clarified this point in the corresponding Results section.

The observation of larger focal adhesions in the center of cells expressing vinculin mutants does not seem to me to demonstrate the existence of a maturation phenomenon of vinculin that would strengthen its link with talin over time. I suggest to the authors that an interesting experiment would have been to measure the dissociation rate of WT vinculin by FRAP in increasingly older focal adhesions (from the leading edge to the center), i.e. containing talin-vinculin complexes with increasingly longer lifetimes.

The association with central adhesions is not reflecting maturation directly, but, more simply, suggests that the central adhesions can effectively interact with the mutant molecules and not (or a lot less) with WT vinculin. It has been shown long ago that central adhesions, such as fibrillar adhesions, are not affected by blocking acto-myosin tension (Zamir et al, Nat Cell Biol, 2000). In our study, we further show that the central adhesions lack zyxin, which is known (by us and by others) to localize at 'high tension adhesions'.

The FRAP experiments are an interesting suggestion. We in fact in an earlier study compared central and peripheral FAs and the lifespan of FA components in this FAs by FRAP after relaxation by contraction inhibitors (Lavelin et al, doi:10.1371/journal.pone.0073549). Mechanical relaxation leads for virtually all FA molecules to recoveries which are slower at peripheral as opposed to central adhesions, suggesting stronger associations overall. We would hesitate to interpret this in terms of vinculin maturation, as this seems to result from overall differences in the assembly for different types (or ages) of FAs.

We now cite both studies mentioned above in the corresponding Discussion section.

The novelty of the conclusions derived from figures 1, 2 and 3 is limited as indicated by the titles of the figures. Fig.1 : Vinculin's head-tail interface regulates vinculin-VBS binding ; Fig.2 : Talin binding facilitates vinculin head-tail opening ; Fig :3. Talin binding weakens the vinculin head-to-tail interaction. These conclusions confirm what is already documented and well established in the field by a large amount of publications that are unfortunately not all cited here. The methodology which consists in putting under tension single molecules of talin and vinculin by magnetic tweezers has also been used in the past on several occasions by several laboratories. Moreover, this setup has been used by "Wang et al. Force-Dependent Interactions between Talin and Full-Length Vinculin. J Am Chem Soc. 2021" to determine that mechanically

exposed VBS binds vinculin significantly more tightly than the isolated non-stretched VBS alone, which is also the conclusion of the present study by Franz et al. However, I agree that the concept of a vinculin maturation that reinforces its link with talin after binding is a new and interesting discovery.

We thank the reviewer for appreciating our discovery as new and interesting. Regarding previous single-molecule works studying vinculin-talin interaction under force, and the mentioned JACS paper in particular, our work presents several novel aspects that were not assessed there, and also some contradicting results. First, our main finding is the identification of an allosteric vinculin activation mechanism as a maturation transition triggered upon binding to talin, related to a weakening of the head-to-tail interaction. Second, we do not conclude that vinculin binds more tightly to stretched VBSs than to non-stretched ones; oppositely, we show that vinculin is able to bind talin more tightly at lower forces, since the maturation rate outcompetes the unbinding rate. Third, our results agree with those of (Wang et al JACS 2021) in that full-length vinculin binds more weakly talin than the D1 domain alone (lower affinity). However, we show that this is because of the faster unbinding kinetics and not due to the binding rates, which are indeed equivalent. Finally, from a technical perspective, our work shows two improvements compared to (Wang et al JACS 2021): 1) In our single-molecule assay, we directly monitor the dynamics of vinculin binding by resolving individual vinculin binding events over very long timescales, which allows us to identify the maturation mechanism, while Wang et al study vinculin binding indirectly, simply inferring the bound/not-bound state over a fixed time window. 2) We used vinculin expressed in an eukaryotic system, and therefore with its physiological post-translational modifications (PTMs), while Wang et al used bacterial expression that do not introduce these PTMs. For these reasons, we believe our work significantly advances our knowledge of the force-dependent talin-vinculin interaction, in particular uncovering a molecular mechanism for vinculin activation.

The 5M mutant of vinculin is novel and will likely be useful in mimicking the active conformation of vinculin. However there are already several active vinculin mutants such as the most widely used called T12. The bundling activity of vinculin has also been reported several times and it is already accepted that this activity is carried by Vt exposed following vinculin activation. In the context of the present study, it is not clear what the use of this mutant in cells or in bundling assays add to the demonstration of a maturation process of vinculin. Other studies have shown that vinculin activation promotes focal adhesion formation and actin binding.

We now have conducted new single-molecule and actin-bundling experiments to evaluate the binding of the vinculin-T12 mutant to talin R3 under force and its bundling efficiency in absence of VBS (new Figs. S9 and S10). In both sets of experiments, T12 shows an overall similar behavior to the 4M and 5M mutants. Also, we now refer to previously published live cell experiments with T12, which also show more central adhesions. Taken together, this corroborates that the 4M and 5M mutants are in an opened conformation. In fact, T12 more resembles 4M than 5M in its capacity to bundle

actin. 5M thus is a more effective bundler in vitro. It is important to note that while the T12 mutations were designed to directly weaken the head-to-tail interface, the 4M and 5M mutants abrogate the allosteric-network involved in talin-induced vinculin activation. Thus 4M/5M here serve as a validation of the allosteric mechanism identified by Molecular Dynamics simulations.

We have added the new T12 data to the SI (Figs S9/S10), refer to this new data in the respective Results sections, and discuss our M4/M5 mutants in context of this previously studied T12 mutant in the Discussion.

In the discussion section, many references should be cited for the following sentence "Actomyosin-dependent forces, acting on vinculin are not required but can, of course, further enhance the maturation by shifting the vinculin conformational ensemble further to the extended open state of vinculin."

We added references for this statement. We emphasize that the references refer to the fact that forces on vinculin are not required for talin binding, while the novelty here lies in the maturation.

Although the technical quality of this study is excellent and the discovery of vinculin maturation is interesting, it must be said that the novelty of the concepts presented is somewhat limited for a wide audience.

The mechanism of vinculin activation has so far remained elusive. We here show that an allosteric network connects the talin binding site of vinculin to the head-tail interface and allows maturation. We see two major findings to be of high interest to a wide audience: (i) the existence of a maturation process as a vinculin activation mechanism and (ii) its consequence to the mechanical forces involved in focal adhesion assembly, arising from the competence between vinculin activation and unbinding. We therefore believe that our findings bring key novel insights into the yet unclear mechanism of vinculin activation and, more broadly, demonstrate the intricate role of mechanical forces in regulating molecular behavior. Moreover, provided our multiscale approach (spanning from atomistic simulations to single cell experiments, with high resolution single-molecule experiments and actin bundling assays in between), our results resonate with a broad scientific community, specially within the burgeoning field of mechanobiology.

We added a Conclusion section to stress the novelty of our findings.

Reviewer #2 (Remarks to the Author):

In the manuscript “How talin allosterically activates vinculin” the authors describe a maturation mechanism between vinculin and force-activated talin. Full length vinculin forms a less stable complex with talin than the isolated (vinculin-binding) D1 domain, as the authors convincingly show that the head-tail internal interaction of vinculin weakens its external interaction with vinculin binding sites (VBS) fragments through an allosteric mechanism. This finding supports a mechanism where a more stable complex between VBS and vinculin requires its head to be engaged in a competing interaction with actin. Interestingly, the authors find that a maturation mechanism taking place after a few seconds and at intermediate forces, can strengthen the interaction of vinculin with the VBS from talin. Equally important, full-length vinculin, which is known to be in an autoinhibited form, seem to be able to bind to exposed vinculin binding sites (VBS). The authors then use molecular dynamics simulations to describe the mechanism in detail and find two mutants that would enhance the vinculin-VSB interaction.

Major comments:

1. In figure 1 a comparison is made between the response to force of the VBS fragment in the presence of vinculin head and full length vinculin. The authors seem to be missing a critical control. What happens when the VBS fragment is pulled without vinculin being present? They should add those passage times to Figure 1c.

The VBS fragment is a helix which is highly unstable under force. At the force range shown in Figure 1c, it would be fully extended, and it needs the stabilization through vinculin interactions to be mechanically resistant. We performed MD simulations of VBS1 under different constant forces, and observed the complete loss of helicity beyond ~10pN (Figure R1).

Figure R1: (a) VBS1 end-to-end distance and (b) helicity when subjected to forces ranging from 0 to 20pN. For different force fields, the standard deviation (shade) and average (solid lines) are shown.

Furthermore, we recently demonstrated that talin R3 has a rich conformational space, populating several low-probability states (*Tapia-Rojo et al Nat Phys 2023*). One of such states corresponds to the spontaneous reformation of the VBSs helices (Fig. R1). However, this state appears with very slow kinetics ($\sim 0.0008 \text{ s}^{-1}$) and has a fast escape rate ($\sim 0.01 \text{ s}^{-1}$), suggesting that, in the absence of vinculin, the VBSs helices even under low forces ($\sim 9 \text{ pN}$) are very unstable and uncoil quickly, in agreement with the SMD simulations.

Figure R2: Spontaneous reformation of the VBSs helices in talin R3 at 8.5 pN in the absence of vinculin.

Given that the instability of the VBS-helices under force is a somewhat expected result, and that our data shows that it occurs over a completely different force range and timescale, we have decided to only include the data as figures for the reviewer (Figs. R1 & R2) not to unnecessarily increase the length and complexity of our manuscript.

2. Do the two Ig32 domains unfold at higher forces during the magnetic tweezers experiments, when exposed to higher forces? Could they constitute a fingerprint for this molecular system?

The titin Ig32 domains show no response over the explored range of forces (<40 pN), given their high mechanical stability, requiring forces in excess of 100 pN to be unfolded (see Fig. R3). Therefore, we use them as molecular handles to keep the protein of interest away from the surface and bead. They are not an adequate fingerprint system due to the high forces required for unfolding, which would frequently lead to the rupture of the non-covalent biotin-streptavidin interaction. Moreover, the folding dynamics of talin R3 have been shown to be independent of the employed molecular handles (see *Tapia-Rojo Nat Phys 2023*).

Figure R3: Unfolding of Ig32 using magnetic tweezers. (A) Schematics of the protein construct, comprising four Ig32 domains (purple), flanked by a HaloTag for covalent attachment to the glass (yellow) and biotin (green) for interaction with the superparamagnetic bead. (B) Magnetic tweezers recording showing that Ig32 unfolds at 100 pN over several hundredths of seconds, and does not refold at 8 pN. Therefore, Ig32 is a good molecular handle for the study of proteins with a low mechanical stability such as talin. Figure adapted from SI of *M. Mora et al. Nano Lett. 21, 2953–2961, (2021)*

3. The authors state that their setup with the magnetic tape operates in the 0-40 pN range. However, the data in figure 1G shows unbinding rates at 60 pN. Are those rates coming from the magnetic tweezers experiments or from simulations? If not, clarify this discrepancy between the working range of the magnetic tweezers and the applied force in Fig. 1G.

All experiments for this manuscript were conducted on our tape head magnetic tweezers setup (*Tapia-Rojo, PNAS 2019*), which allows applying forces up to ~40 pN. The experiments on the D1 domain (*Tapia-Rojo Sci Adv 2020*) were also conducted on a different setup that uses permanent magnets, which allows for forces up to ~110 pN.

4. The fast part of the double exponential relies on fitting a single bin, which probably does not produce a statistically significant value, as is dominated by the size of the bin used to make the histogram. Maybe a histogram of the logarithm values of the measured unbinding times can produce more accurate values on the fast and slow unbinding rates.

We have now calculated the histogram with logarithmic binning, which, as suggested by the reviewer, allows for a more accurate identification of the involved timescales. See also our answer to a similar question from reviewer #1.

5. The dual behavior for unbinding of vinculin when attached to talin is a very interesting result. The authors find that the vinculin domain binding talin (Vd1) and full-length vinculin (WT) have similar unbinding kinetics from talin R3 domain, supporting the idea that this dual behavior is not related to the direct interaction between talin and vinculin. The authors propose that vinculin can undergo a slow conformational change when bound to a VBS of talin and even propose two mutants and a molecular mechanism for it. However, in their simulations the VBS fragment does not seem to be under force. It was recently shown that mechanical activation of talin R8 can induce mechanically stable states when ligands bind. Is this the case for the VBS site? Simulations with this fragment under a force vector could give some insight.

This is an interesting question. We did simulate the VBS-vinculin system with VBS under pulling force and found the presence of the vinculin tail to destabilize the complex, more specifically, to lead to more rapid VBS unfolding and dissociation at a given external force (Figs. 1A/B, Fig. S1). We explain this by a competition of the interactions of VBS and the tail with D1 (Fig. S2). In these simulations, we consider vinculin to be in the matured state, as we start from the crystal structure with VBS bound, and the time scale of maturation is such that it can not be covered by atomistic MD simulations. However, our aggregated data from simulations and experiments support the notion of mutual force-induced strengthening we believe the reviewer is pointing at: VBS binding in mechanically activated R8 of talin leads to maturation, and by loosening of the head-tail interface, this maturation in turn strengthens the binding to activated/stretched talin. We have added this to the Discussion section.

6. In a different article, also not cited here, vinculin-VSB interaction was shown to have a reinforcement mechanism occurring along preferential force directions ([10.1016/j.bpj.2019.12.042](https://doi.org/10.1016/j.bpj.2019.12.042)). This article seem to also propose a time and force dependent binding mechanism. Can the authors comment on how their findings are similar/different?

We have been well aware of this related and important work and now cite it in the Discussion. In this study, the authors probed the mechanical response when pulling VBS (at N- or C-terminus) away from vinculin's D1, i.e. along a force direction orthogonal to the one employed here in experiments and simulations (force between the two VBS termini). Thus, the authors in the previous study focused on the mechanobiology of the system after vinculin is fully activated while being tethered on both sides. Instead, we here concentrate on the preceding step, the initial recognition and activation of vinculin by talin.

7. The model proposed here (Figure 3F) claims that the stable state is a force activated state that requires a low force (~ 8 pN) and would not exist under a high force (~15 pN). Do the MD simulations measured both of these states in a force-dependent manner?

The reason for the stable state existing only at low but not at high forces is the competition between maturation into the stable state and dissociation (with the latter dominating at higher forces). For the matured state we do not have an atomistic model, but for the fully activated state (Vd dissociated), we indeed find a higher mechanical stability (Fig. 1C), supporting the notion that maturation destabilizes the Vh-Vt interface. Remarkably, we could also recover in MD simulations the 'non-matured' state of inhibited vinculin (with a strong Vt-Vd interface) from the VBS-bound state (with a weak Vt-Vd interface, Fig. S5). We now refer to these two findings when describing the model of Fig. 3F in the Results section.

8. Furthermore, to confirm the hypothesis mentioned at the points above, the authors performed single molecule measurements of talin unbinding in the presence of vinculin mutants and see a significant change in the unbinding time of these mutant when compared to WT vinculin (Figure 4F). The times reported in this figure for WT (~10 s) and Vd1 (~100 s) seem to however disagree with the statement that the authors make, that the unbinding kinetics of Vd1 and WT are similar (first and last bar in Figure 4F). Please explain this difference.

The differences were likely due to poor statistics in our initial manuscript submission. Motivated by this reviewer's suggestion, we have now conducted more experiments on the 4M and 5M mutants to better quantify their unbinding kinetics, which have also allowed us to calculate the distribution of unbinding times for both mutants. As shown in Fig. 4D-G, both mutants display a single bound mode (single-exponential distribution of unbinding times), and show no statistically significant differences in their unbinding kinetics. This suggests that the 4M and 5M mutations lead to a similar pre-active vinculin that binds tightly to talin.

9. Also the authors claim that the 4M mutant does not undergo any maturation process. However the rate measured for unbinding of this mutant (Figure 4E – showing 4M mutant that does not undergo the maturation process) is much slower than the maturation rate for the WT vinculin (Figure 3C). These results suggests to me that the mutant rather undergoes maturation at much faster rate and the measured state is rather related to this 'mature' state.

We agree with the reviewer that both the 4M and 5M mutants could undergo a very fast maturation process that we are not able to capture with our experimental resolution. In our experiments, we unbind vinculin by applying a 40 pN pulse after observing a binding event. Therefore, there is a natural limitation in how quickly this pulse can be applied, simply because the experimentalist needs to observe the binding event and manually change the force to trigger vinculin unbinding, which can take at least ~3-5 s. This means that if the 4M and 5M mature very quickly upon binding to vinculin (<3 s), we will be only capturing the population of mature states. Regardless of whether this is the case or if the

4M and 5M mutants directly lead to an open vinculin configuration, our data indicates that these mutations weaken vinculin's head-to-tail interaction, either by directly abolishing it or by leading to vinculin molecules that can open very quickly. We have clarified this now in our manuscript.

10. The microscopy based data seems to contradict the single molecule measurements, with one showing that mutant 4 M forms a more stable complex than 5M, while the other the opposite. How do the authors reconcile these results?

5M lacks an additional interfacial salt bridge by the E986A mutation, and thus was intended to show a stronger VBS-simulating effect. Indeed, the bundling data shows a higher bundling activity for 5M compared to 4M. The single molecule data, however, does not show a significant difference in the 4M and 5M affinities, suggesting that both variants form a similarly stable complex with talin. We also would like to note that the two sets of experiments must not result in data that quantitatively correlates, as the single molecule data measures vinculin dissociation from VBS while the in vitro assembly measures actin bundling activity. In the Discussion section, we focus on the mutants' different activities in actin bundling, as this is the more functional readout.

11. What is the scale of the images in Figure 6A? Also the font size used in Figure 6B-E is too small to be easily read.

The scale bar is 20 μ m, we added this information. Fig. 6 was updated and now has a larger font.

12. Several single molecule studies have shown so far a binding-induced stabilization effect, both on the proteins reported here, as well as other proteins. A small discussion on this topic and comparison with those studies would greatly benefit this manuscript.

We thank the reviewer for bringing this topic of discussion, which we believe is very relevant and interesting. As the reviewer highlights, there are several single-molecule works that study the binding-induced stabilization effect on a protein under force. In the context of mechanotransduction, vinculin is known to stabilize the unfolded conformation of both talin and alpha-catenin (*M Yao Nat Comm 2014*), but other talin ligands have been shown to stabilize the folded state of talin, such as DLC1 binding to the talin R8 domain (*N. Dahal Sci. Adc. 2022*). More broadly, this seems to be a more general physical effect to folding/unfolding proteins; the trigger factor chaperone was shown to bind to unfolded proteins to decrease their entropy and favor the folding transition (*Halder Nat Comm 2017*), while protein L stabilizes upon antibody binding (*N. Dahal J Phys Chem B 2020*). However, our findings here do not relate to the impact that binding has on the substrate molecule (it was already known that talin is locked in the unfolded state upon vinculin binding), but on the effect the interaction has on the protein binder (here, vinculin changes conformation upon binding). Therefore, we believe this is a novel discovery that highlights the intricate relationship between mechanical forces,

protein folding, and binding interactions, of particular relevance in the context of mechanotransduction.

We have now expanded the discussion to comment on this topic in light of the different literature.

13. The magnetic tweezers data is of very high quality and provides the strongest support for the conclusions of the paper. In the main text Figure 1, 3 and 4 rely on these data. According to author contributions, those measurements were performed by the second author, who in my opinion deserves at least equal contributor/co-first author status, given the amount of work he put in compared to the other measurements reported in this manuscript.

We fully agree and this is now reflected by a shared first authorship of the two first authors that have performed the MD simulations and tweezers experiments, respectively.

Minor comments:

According to Merriam Webster dictionary, the noun “tweezers” is “plural in form but singular or plural in construction”. Correct the abstract where it appears as “magnetic tweezer” to read “magnetic tweezers”, which is also the correct name for this single molecule technique. On the other hand, also in the abstract, “two-ways network” should be “two-way network”.

We thank the reviewer for spotting this; we have now corrected it.

The authors could probably use the same 20 s scale bar in both panels of Figure 3.

We thank the reviewer for noticing this. We have fixed it.

Page 1 – the part of the second sentence starting with “when the tiny ...” seems to be missing the verb

This was fixed

Page 2 – remove the comma between “10” and “pN”

This was fixed

Page 2 – replace “composed of 1066-residue-long” to “composed of 1066 residues”. Also check that indeed vinculin has 1066 amino acids. Uniport.org reports 1134 amino acids for human vinculin.

We agree that Uniport.org reports 1134 amino acids for human vinculin, which has three isoforms produced by alternative splicing. Metavinculin is the P18206-1 isoform that has been chosen as the canonical sequence, and is 1134 amino acids long. Vinculin, which we used in our study, is the P18206-2 isoform lacking 68 amino acids (916-983), and is indeed 1066 amino acids long. Metavinculin is muscle-specific and is co-expressed with vinculin in cardiac tissue and smooth muscles. Vinculin is expressed ubiquitously.

Page 4 – clarify what you mean by “actin is not able to bind to on its own”
We rephrased and now state “actin is not able to bind to inactive vinculin.”

Page 5 – under “Results” second and third sentences need some rephrasing or are missing words
We rephrased the sentences.

Reviewer #3 (Remarks to the Author):

The paper by Gräter et al. presents an important combined experimental and MD simulation work resulting from the cooperation of several groups. The activation mechanism of vinculin and interaction with talin is studied, and fundamental results for actin bundling and cell association are obtained.

We thank the reviewer for appreciating the importance of our study.

Unfortunately the quality of presentation is by no way adequate to the reported results, please regard the comments I made to the original manuscript in the appended file:

- The text contains very confusing unclear phrases.

We thank the reviewer for the detailed comments in the manuscript, and revised the text accordingly.

- Important details on numerical analysis (Bell-Evans, dose-response) and simulation have not been given.

We added analysis details to the revised version wherever possible.

- The important results are obscured by poor writing, the need of presenting many details and the lack of a conclusion section.

We thoroughly rewrote the decisive sections in the Results and Discussion and added a Conclusions section to clearly state the major points we make.

The resulting manuscript is neither an instructive short communication of important findings nor a satisfying self contained report on their work.

I suggest that the authors first publish a letter focusing on the most important conclusions and then a really complete report elsewhere.

We believe that a joint publication of MD simulations, single-molecule data, the in vitro bundling results and cell experiments is required to support our conclusion of vinculin maturation as an allosteric consequence of talin binding.

Fig. 1 Bell-Evans' model: The application of the Bell-Evans model has to be explained and the parameters used have to be explained, and reference to literature has to be made. 1/MFPT

seems to be derived from the loading rate. Obviously k_0 and x_t are related to intercept and slope of the straight lines, respectively. The quality of the data both in Figs 1C and 1G is low, and there is no reason to use different x_t for the both parallel lines in Figs 1C (0.25 and 0.23 nm) and 1G (0.72 and 0.81 nm). Common parameter for x_t (e.g. 0.24 and 0.75 nm, respectively) would make the interpretation of results for k_0 more sensible.

We thank the reviewer for pointing this out. The Bell-Evans model is the simplest model for describing the force-dependence of a reaction-rate, such as for protein unfolding or, as in this case, protein-protein unbinding, and it has been widely employed over the last few decades. We have included references to justify its use in the methods section. Figure 1C shows the inverse of the 1/MFPT which, in the case of a single kinetic process, is equal to the reaction rate (see e.g. *Hänggi Rev. Mod. Phys.* 1990). Therefore, Fig. 1C and Fig. 1G show the same physical magnitude: the unbinding kinetics of vinculin. We are unsure what the reviewer means by “the quality of the data both in Figs 1C and 1G is low”. The unbinding rates have been measured from a large number of events (more than 700 single-molecule events for Fig. 1G), which provides a solid quantification of the vinculin unbinding kinetics. The dashed lines in Fig. 1C and 1G correspond to fits of the Bell-Evans model to each data set; hence, we cannot assume that their slope is the same. Precisely, from the fit, we can conclude that the x^\dagger for both the D1 and full-length vinculin domain is similar within errors, which suggest that the differences in the unbinding kinetics are in the k_0 value.

We have now explained the employed model in the SI and included a table with the obtained parameters in the SI (Table S3).

Fig 3c: For my understanding the statistics of these data are definitely too poor for deriving a double exponential function.

We have now calculated the distribution using a logarithmic binning, which allows identifying two involved timescales as two peaks in the histogram; one for the weak state, and one for the mature state. See also reply to Reviewer #1.

Fig. 5D: “four-parameter dose-response equation” What function did you use, and what are the four parameters? There are many possibilities to generate sigmoid functions. Give reference for the Graphpad software.

We added this information to the respective Methods section and now refer from the Figure legend of Fig. 5D to the Methods.

Fig 6: “Naturally, the non-transfected (Control) cells were GFP-negative, yet they displayed poorly-organized β_1 integrin, lacking the typical morphology of FAs.” Difficult to understand: If you detect β_1 integrin in transfected cells by GFP, how can you find “poorly organized β_1 integrin” in the GFP-negative control cells?

To clarify, GFP labeling was used for vinculin and its mutants, and b1 integrin was detected not by GFP but by antibodies. b1 is poorly organized in the controls, while zyxin was associated with FA-like structures (possibly via interaction with another integrin (e.g. avb3). We clarified the detection of b1 in the Results section.

“A large body of work has confirmed the need of talin to be activated for vinculin by VBS exposure.” Citations?

We have added a number of references.

“(note that in contrast to this previous study using vinculin from bacterial expression, we here used eukaryotic expression with post-translational modifications in place).” Does this mean that your vinculin was possibly corrupted by later treatment and had other properties than in vivo? This could put in question your experimental results!

We believe this is precisely the opposite. Vinculin in vivo is post-translationally modified; therefore, the eukaryotic expression leads to vinculin with the same “chemistry” as in vivo and, allegedly, the same behavior. If such PTMs are relevant for the binding behavior of vinculin, studies using vinculin from prokaryotic expression could potentially lead to artifactual observations.

“The approximate cell radius was then defined as $r = \sqrt{\text{area}/2\pi}$.” Where does the factor of 2 in the denominator come from? For a circle cross section, r would be root of area/π ?

This was a typo and we apologize for the error. We removed 2 from the equation.

REVIEWERS' COMMENTS

Reviewer #1 (Remarks to the Author):

The authors performed new magnetic tweezers experiments to study the binding of full-length vinculin to talin R3WT, as requested, and reached the same findings as for R3IVVI (new Figs. S4 and S6). They also calculated and plotted the dwell-time distribution with logarithmic binning (Fig. 3C) to demonstrate the existence of distinct populations in the distributions. As requested, new single-molecule and actin-bundling experiments are presented to measure the binding of the vinculin-T12 mutant to talin R3 under force and its actin bundling activity in absence of VBS (new Figs. S9 and S10). The behaviour of T12 is similar to that of the 4M and 5M mutants. Several clarifications have also been added in the results and discussion sections.

As the majority of the points raised in my previous review have been satisfactorily addressed, I consider that this work is now acceptable for publication.

Reviewer #2 (Remarks to the Author):

The authors have adequately answered all the points raised.

Reviewer #3 (Remarks to the Author):

The manuscript has been considerably improved by adding several explanations. I still want to make these remarks:

- The expression "square-root histogram" is not obvious to me. The authors obviously use as basic function $f(t)=t/t_0*\exp(-t/t_0)$ which probably is the solution of any kinetic equation not mentioned in the text, but where does a "square root" come from?
- The fits by a dose-response model in Fig 5c essentially describe sets of three to five data points by four (!) independent parameters for each. This will work with essentially any type function and does not validate the underlying model at all. The occurrence of R^2 values close to 1 is trivial for these fits.
- If you go for EC50, a Hill plot is more instructive.
- The Hill equation " $Y = b + (X^{\text{Hillslope}}*(a-b)/(X^{\text{Hillslope}} + EC50^{\text{Hillslope}})$ " should be rewritten with a proper formula editor, even wikipedia can do better (https://en.wikipedia.org/wiki/Dose%E2%80%93response_relationship).
- The equations in the section "Analysis of vinculin.." and "Square root.." are not properly reproduced in the pdf and seem to contain some blanks even in the docx, please check this.
- Equations should be numbered and appropriately referenced.

I still think that the paper is too long for a "communication", but it should be ready for publication with small improvements.

Reviewer #1 (Remarks to the Author):

The authors performed new magnetic tweezers experiments to study the binding of full-length vinculin to talin R3WT, as requested, and reached the same findings as for R3IVVI (new Figs. S4 and S6). They also calculated and plotted the dwell-time distribution with logarithmic binning (Fig. 3C) to demonstrate the existence of distinct populations in the distributions. As requested, new single-molecule and actin-bundling experiments are presented to measure the binding of the vinculin-T12 mutant to talin R3 under force and its actin bundling activity in absence of VBS (new Figs. S9 and S10). The behaviour of T12 is similar to that of the 4M and 5M mutants. Several clarifications have also been added in the results and discussion sections. As the majority of the points raised in my previous review have been satisfactorily addressed, I consider that this work is now acceptable for publication.

We thank the reviewer for their positive review.

Reviewer #2 (Remarks to the Author):

The authors have adequately answered all the points raised.

We thank the reviewer for their positive review.

Reviewer #3 (Remarks to the Author):

The manuscript has been considerably improved by adding several explanations. I still want to make these remarks:

- The expression "square-root histogram" is not obvious to me. The authors obviously use as basic function $f(t)=t/t_0 \cdot \exp(-t/t_0)$ which probably is the solution of any kinetic equation not mentioned in the text, but where does a "square root" come from?

A square-root histogram is simply a logarithmic transformation ($x=\ln t$) of an exponential distribution. When plotting with a logarithmic binning, the timescale of the exponential appears as a peak in the distribution ($x_0=\ln t_0$), and hence, kinetic processes involving more than one timescale are more easily to identify provided the involved timescales are well-separated. Reference *FJ Sigworth Biophys. J 1987* provides more detail on this method for identifying multiple kinetic processes on experimental histograms. The term "square-root histogram" is due to the normalization of the bins' counts. We have now explained in more detail this transformation in the Methods section.

- The fits by a dose-response model in Fig 5c essentially describe sets of three to five data points by four (!) independent parameters for each. This will work with essentially any type function and does not validate the underlying model at all. The occurrence of R^2 values close to 1 is trivial for these fits.

We apologize for being unclear in describing the procedure used for fitting curves into our results. The fit of the overall dataset was obtained with a dose-response model using the Hill- Langmuir equation, where the best-fit values for the four parameters, plateaus, Hill slope and EC50 were calculated from the overall dataset, with the value of the saturation plateau to be shared between all datasets. We now edited the Methods section accordingly.

In the revised version, we have integrated the data representing the effect of vinculin T12 into the initial dataset, as shown in Fig S10D. This had only a negligible effect on the best-fit values calculated initially. We also added to the supplementary information a table summarising the best-fit values.

Moreover, we represented actin bundling by vinculin variants by plotting the ratio of bundled actin as a function of the concentration of VBS1. This reflects the occupancy of the substrate, here actin filaments by vinculin, and displayed a sigmoidal dependence of the “binding and simultaneous bundling” (as previously shown by Boujemaa-Paterski et al, eLife, 2020) of the actin. Since the determination of the dissociation constant of the observed reaction may reveal the molecular complex (i.e. vinculin activation by VBS, subsequent binding and bundling of decorated filaments), we used the half maximal effective concentration which describes the overall outcome.

- If you go for EC50, a Hill plot is more instructive.

We thank the reviewer for this comment. As vinculin variants have a variable ability to bundle actin filaments in the absence of VBS, the logarithmic scale in the Hill plot represents less the sigmoidal dependence of the bundling than the decimal scale. In any case, we now provide a table summarising the best-fit values, including that of Hill slope, in the supplementary information.

- The Hill equation " $Y = b + (X^{\text{HillSlope}})^*(a-b)/(X^{\text{HillSlope}} + EC50^{\text{HillSlope}})$ " should be rewritten with a proper formula editor, even wikipedia can do better (https://en.wikipedia.org/wiki/Dose%20%80%93response_relationship).

We have followed this reviewer’s suggestion and rewrote the formula with the equation editor

- The equations in the section "Analysis of vinculin.." and "Square root.." are not properly reproduced in the pdf and seem to contain some blanks even in the docx, please check this.

We thank the reviewer for noticing this. We have corrected it accordingly.

- Equations should be numbered and appropriately referenced. I still think that the paper is too long for a "communication", but it should be ready for publication with small improvements.

We have numbered the equations following the reviewer’s suggestion.